# Transient microglial absence assists postmigratory cortical neurons in proper differentiation

Yuki Hattori [1,2✉], Yu Naito[1], Yoji Tsugawa[3,4,5], Shigenori Nonaka [6,7], Hiroaki Wake [8,9,10,11], Takashi Nagasawa[12], Ayano Kawaguchi [1] & Takaki Miyata [1✉]

In the developing cortex, postmigratory neurons accumulate in the cortical plate (CP) to properly differentiate consolidating subtype identities. Microglia, despite their extensive surveying activity, temporarily disappear from the midembryonic CP. However, the mechanism and significance of this absence are unknown. Here, we show that microglia bidirectionally migrate via attraction by CXCL12 released from the meninges and subventricular zone and thereby exit the midembryonic CP. Upon nonphysiological excessive exposure to microglia in vivo or in vitro, young postmigratory and in vitro-grown CP neurons showed abnormal differentiation with disturbed expression of the subtype-associated transcription factors and genes implicated in functional neuronal maturation. Notably, this effect is primarily attributed to interleukin 6 and type I interferon secreted by microglia. These results suggest that "sanctuarization" from microglia in the midembryonic CP is required for neurons to appropriately fine-tune the expression of molecules needed for proper differentiation, thus securing the establishment of functional cortical circuit.

[1] Department of Anatomy and Cell Biology, Graduate School of Medicine, Nagoya University, Nagoya, Japan. [2] Japan Society for the Promotion of Science, Tokyo, Japan. [3] Department of Aging Intervention, National Center for Geriatrics and Gerontology, Obu, Japan. [4] Laboratory of Molecular Biotechnology, Graduate School of Bioagricultural Sciences, Nagoya University, Nagoya, Japan. [5] Drug Discovery Research, iBody Inc., Nagoya, Japan. [6] Spatiotemporal Regulations Group, Exploratory Research Center on Life and Living Systems, Okazaki, Japan. [7] Laboratory for Spatiotemporal Regulations, National Institute for Basic Biology, Okazaki, Japan. [8] Division of Homeostatic Development, National Institute for Physiological Sciences, Okazaki, Japan. [9] Department of Physiological Sciences, The Graduate School for Advanced Study, Okazaki, Japan. [10] Division of System Neuroscience, Graduate School of Medicine, Kobe University, Kobe, Japan. [11] Department of Anatomy and Molecular Cell Biology, Graduate School of Medicine, Nagoya University, Nagoya, Japan. [12] Laboratory of Stem Cell Biology and Developmental Immunology, Graduate School of Frontier Biosciences and Graduate School of Medicine, Osaka University, Osaka, Japan. ✉email: ha-yuki@med.nagoya-u.ac.jp; tmiyata@med.nagoya-u.ac.jp

Microglia, the resident macrophages in the central nervous system (CNS), have multiple functions in the adult brain. In pathological contexts, such as neuropsychiatric disorders, neurodegeneration, and infectious diseases, microglia play critical roles as immediate responders with the potential to promote both CNS damage and repair[1–4]. On the other hand, in the healthy brain, microglia contribute to the maintenance of homeostasis by removing dying neurons or cellular debris and monitoring neuronal circuits for successful synaptic connections[5–8].

Embryonic pallial microglia play multiple roles in neurogenesis, e.g., by phagocytotically regulating the number of Tbr2[+] neural progenitors[9,10] and inducing neural stem-like cells to differentiate into Tbr2[+] intermediate progenitors[11,12]. These microglial functions are exerted in the regions in which they exist and are needed. Although microglia account for only a minor population of the cells that constitute the embryonic cerebral wall, they extensively survey the entire structure[12,13] and are thus capable of providing the particular functions that are required in specific regions.

Of note, such actively moving microglia exhibit mysterious behavior in the midembryonic cortex. In mice, intrapallial microglial distribution is initially homogenous until E14, but these cells temporarily disappear from the cortical plate (CP) from E15 to E16 and show preferences for colonizing the ventricular zone (VZ), subventricular zone (SVZ), and intermediate zone (IZ)[9,14]. Considering their contribution to the survival of layer 5 (L5) neurons in the postnatal brain[15], the fundamental question as to why they need to transiently exit the midembryonic CP is raised.

Following birthdate-dependent neuronal specification that occurs in the VZ/SVZ[16,17], postmigratory neurons undergo subsequent differentiation programs that confer their projection subtype identity in the CP[18–20]. For example, SOX5 regulates the differentiation of postmigratory early-born subplate and deep layer neurons[18,19]. The knockdown of protocadherin (Pcdh20) causes the malpositioning of future L4 neurons in L2/3 and induces them to acquire L2/3 characteristics[20]. Microglia indirectly contribute to events in the CP from the outside; intrapallial microglia positioned in the SVZ and IZ regulate the entrance of interneurons generated in the subpallium into the CP and carry out the proper intra-CP positioning of these neurons[14]. However, previous studies have not explored the mechanism and significance of microglial absence from the CP. In light of these observations, we hypothesized that microglia transiently disappear from the midembryonic CP to avoid disturbing neuronal functional differentiation, and we performed experiments to artificially expose CP neurons to excessive microglia.

## Results

**Microglia migrated out from the midembryonic CP.** Microglia are transiently absent from the CP in mouse cerebral walls from E15 to E16[9,14] (Fig. 1a, b; Supplementary Fig. 1). To monitor microglial disappearance from the CP, we live imaged microglia in cerebral wall slices prepared (with intact meninges) from CX3CR1-GFP mice, in which microglia and perivascular macrophages are both visualized[21]. Observation for 8 h on E14 revealed that most microglia initially present in the CP (categorized as those in the outer region of the cortex) migrated toward the basal lamina (basalward migration) (Fig. 1c, d; Supplementary Fig. 2; Supplementary Movie 1). Coimmunostaining for P2RY12 (a specific marker for microglia) and CD206 (highly expressed in macrophages)[22] on E15 revealed that P2RY12[+] microglia existed beneath the meninges, which contained CD206[+] macrophages (Fig. 1e), suggesting that microglia

that migrated toward the meninges accumulated in the marginal zone (MZ). To test whether microglia are attracted by the meninges, we performed live monitoring in cortical slices freed from the meninges. We found that the basalward-predominant migration of CP microglia observed in slices covered with the meninges was strikingly diminished (from 76.7 to 16.7%), concomitant with an increase in the percentage of cells categorized as apicalward migrating (i.e., those migrating toward the apical surface) or stationary (i.e., those that stay in the CP) (Fig. 1f–h; Supplementary Movie 2). The migratory distance of microglia to the meninges was also decreased in meninges-free slices (Fig. 1i). These results suggest that the meninges directly attract microglia in the CP.

To confirm that microglial attraction to the meninges is indeed a physiological phenomenon rather than an artifact of slice culture or surgical procedures, as previously observed in adult brains[23], we established an in utero imaging system using two-photon microscopy by modifying previous methods for the observation of superficially migrating neurons[24,25]. Preparative surgical treatments to mobilize the uterine horn and gently hold the embryo in utero in a device (Fig. 1j, k) enabled us to perform time-lapse imaging for up to 2 h without any issues, such as tilting or horizontal shifts of the field (Supplementary Movie 3), and to monitor microglial migration through a series of z-stack images encompassing the depth of the CP (Fig. 1l). The in utero monitoring of E14 CX3CR1-GFP mice for 90 min revealed that the majority (57.1%) of CP microglia moved basalward, leading to their localization beneath the meninges (Fig. 1m; Supplementary Movie 4). The tendency for meninges-directed microglial movements during in utero monitoring was similar to that in cultured slices (Fig. 1n, o), supporting the model of microglial attraction by the meninges.

We also found that microglia in the inner cerebral wall (VZ/SVZ/IZ) migrated away from the CP and microglia that initially positioned 100–250 μm from the apical surface (corresponding to the IZ) moved more apicalward than cells nearer the apical surface (0–100 μm deep, corresponding to the VZ/SVZ) (Fig. 2a–h; Supplementary Movie 5), suggesting the additional (contradirectional) attraction of microglia to the SVZ/VZ. To study this, the CP or SVZ/VZ explants prepared from E14 cerebral walls were cocultured side by side with CX3CR1-GFP[+] microglia isolated by flow cytometry (FACS) on E14 (Fig. 2i–k). As expected, the density of accumulated microglia in the SVZ/VZ explants after 0.5 day, 1 day, or 3 days was markedly greater than that in the CP explants (Fig. 2l, m; Supplementary Fig. 3).

**The CXCL12/CXCR4 system underlies microglial migration.** To address the molecular mechanism underlying the bidirectional attraction of microglia in the developing cortex, we focused on CXCL12 and its receptor CXCR4, which are involved in microglial recruitment to the SVZ[11]. In situ hybridization demonstrated that *Cxcl12* mRNA was constantly expressed in the meninges starting on E12 and that it tended to gradually increase and was high in the SVZ from E12 to E16, peaking on E14 (Fig. 3a; Supplementary Fig. 4a). Reverse transcription (RT)-PCR revealed *Cxcr4* mRNA expression in CX3CR1[+] microglia (Supplementary Fig. 4b). There were fewer P2RY12[+] microglia in the MZ in cerebral wall slices from CXCR4 knockout (*Cxcr4*[−/−]) mice[26] (Supplementary Fig. 4c), than in those from wild-type (WT) mice on E15 (however, there were comparable densities of P2RY12[+] microglia and CD206[+] macrophages in the meninges) (Fig. 3b–e). This difference in the MZ persisted until E18 (Supplementary Fig. 5a–d) and was reproduced when WT embryos were intraventricularly administered with AMD3100[27], a CXCR4 antagonist (Supplementary Fig. 5e–h). In the CP of *Cxcr4*[−/−]

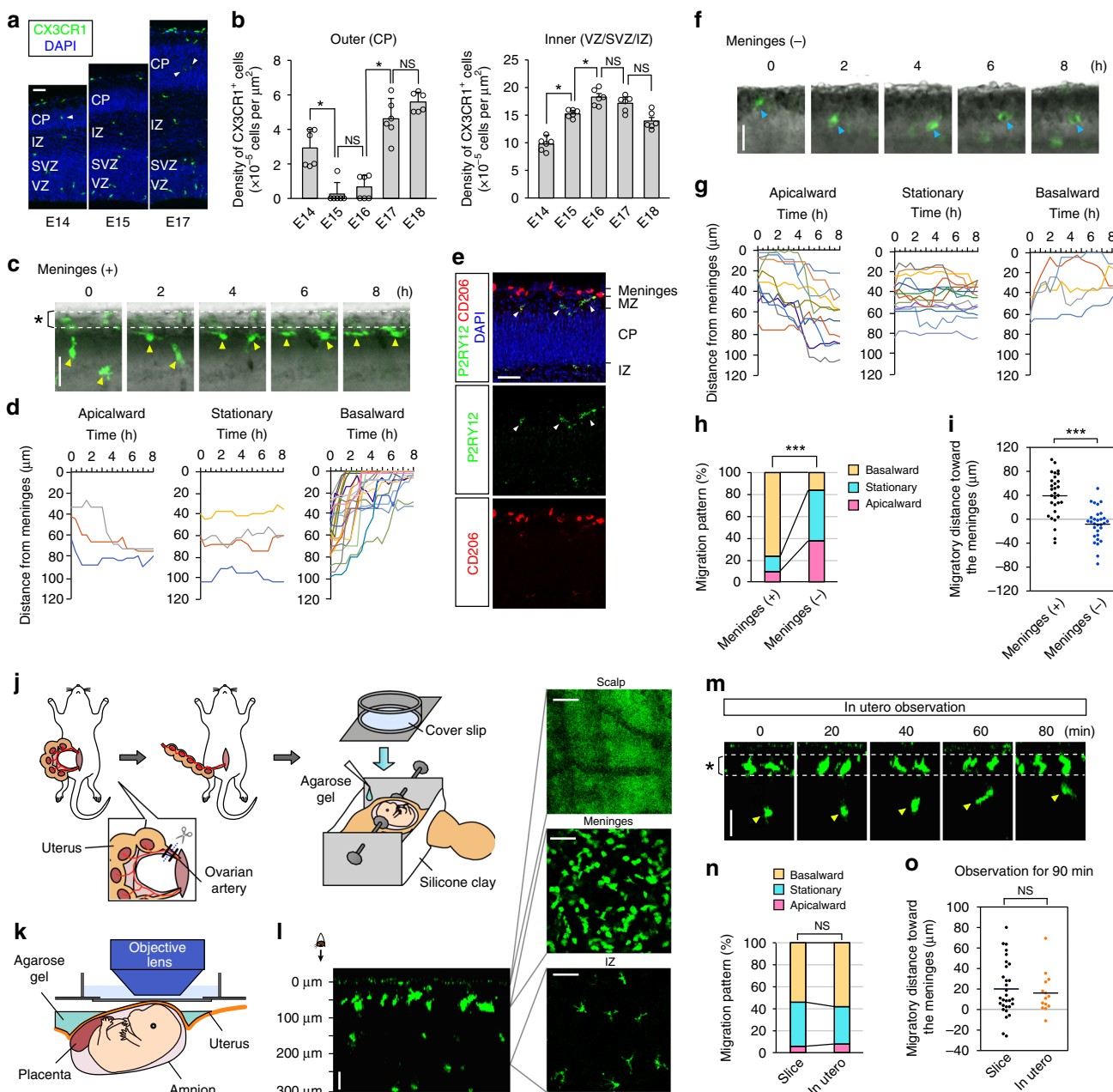

**Fig. 1 Microglia exit the midembryonic CP by meninges-mediated attraction. a** Immunostaining of CX3CR1+ cells in the developing mouse cortex. Arrowhead, microglia in the CP. **b** Comparing the density of CX3CR1+ cells through the stages in the outer and inner region. The density was determined by counting the number of cells inside an area of about 80,000 μm², which covers the dorsolateral cerebral wall in the hemisphere (two-sided Steel–Dwass test; $n = 6$ sections; $P = 0.023$, 0.912, 0.030, and 0.601 for outer, $P = 0.032$, 0.032, 0.703, and 0.115 for inner [left to right]). Data are presented as the mean values ± S.D. **c** Typical migratory behavior of CX3CR1-GFP+ microglia in the CP. Arrowhead, the soma of the microglia. Asterisk, the meninges. **d** The trajectories of microglia that were initially in the outer region (30-min intervals for 8 h) were categorized into three groups (see Supplementary Fig. 2b). **e** Immunostaining for P2RY12 (microglia), CD206 (perivascular macrophages), and DAPI (nuclei) in E15 cerebral walls. Arrowheads, microglia in the MZ. **f** Time-lapse observation of microglia in cortical slices without the meninges. Arrowhead, the soma of the microglia. **g** The trajectories of the microglia in meninges-free slices are shown. A comparison of microglial translocation patterns (**h** two-sided Pearson's chi-squared test; $P = 2.2 \times 10^{-16}$) and the migratory distance toward the meninges (**i** two-sided Mann–Whitney $U$ test; $n = 30$ cells; $P = 5.1 \times 10^{-7}$). A schematic showing the procedure for surgical treatment of the mother mouse (**j**) and the installation of the embryo into the fixing implement (**k**) for in utero observation. **l** Z-stack images. The right panels show the XY-image of the scalp, meninges, and IZ. **m** The in utero observation of microglia that migrated basalward (arrowhead). Asterisk, the meninges. A comparison of microglial translocation patterns (**n** two-sided Pearson's chi-squared test; $P = 0.823$) and the migratory distance toward the meninges (**o** two-sided Mann–Whitney $U$ test; $n = 30$ slices, 14 in utero; $P = 0.775$). Scale bar, 50 μm. Source data are provided as a Source Data File.

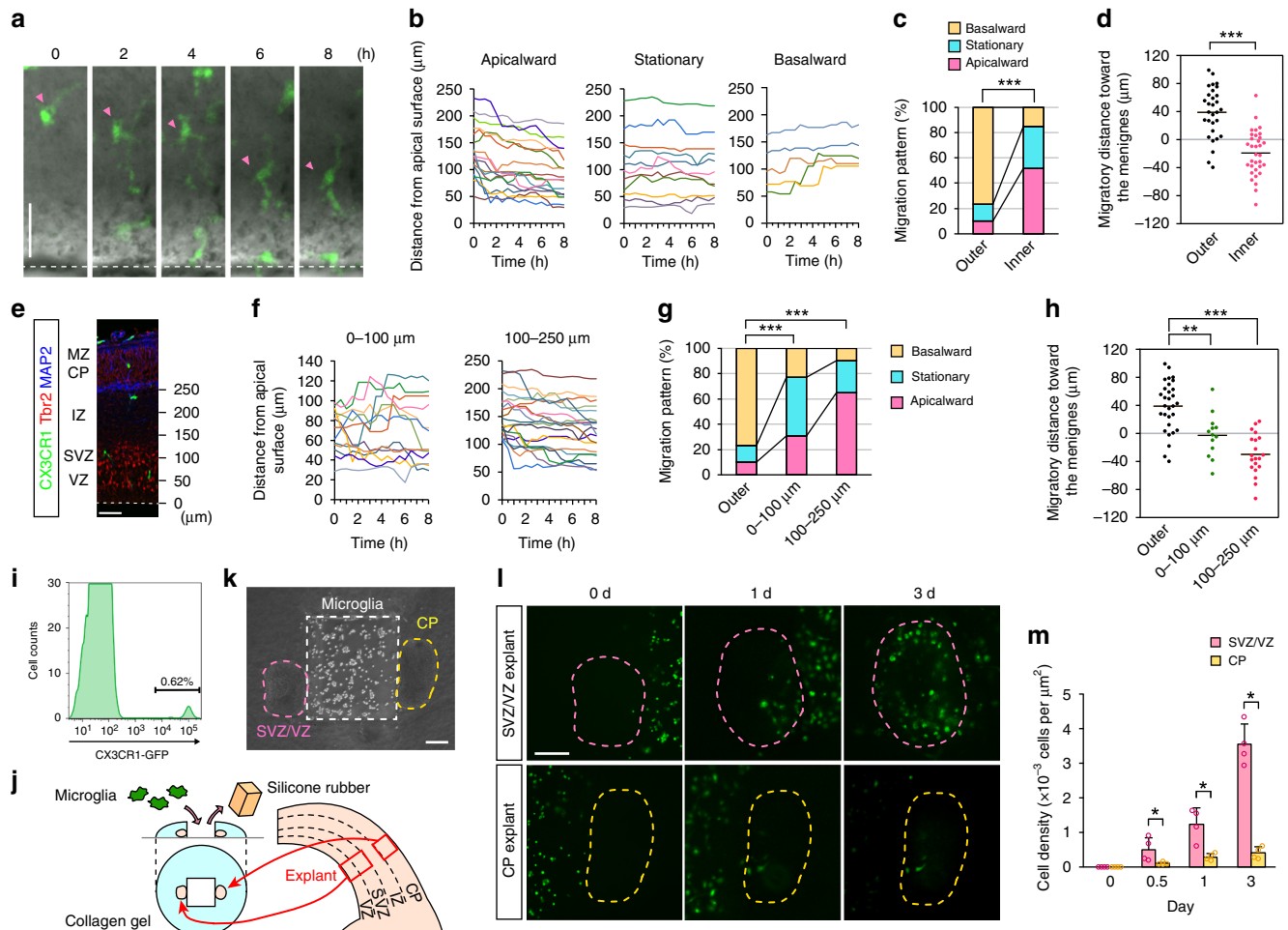

**Fig. 2 Inward attraction from the SVZ/VZ contributes to microglial disappearance from the midembryonic CP. a** Typical migratory behavior of CX3CR1-GFP[+] microglia (arrowhead) that were initially in the inner (VZ/SVZ/IZ) region. Broken line, apical surface. **b** The trajectories of the microglia in the inner region. A comparison of microglial translocation patterns (**c** two-sided Pearson's chi-squared test; $P = 2.2 \times 10^{-16}$) and the migratory distance toward the meninges (**d** two-sided Mann–Whitney $U$ test; $n = 30$ outer cells and 33 inner cells, $P = 3.1 \times 10^{-8}$). **e** Immunostaining for CX3CR1 (microglia), Tbr2 (intermediate progenitors and young neurons positioned in the SVZ), and MAP2 (neurons). Broken line, apical surface. **f** The trajectories of the microglia were categorized into two groups: cells 0–100 μm deep relative to the apical surface and cells 100–250 μm deep relative to the apical surface. A comparison of microglial translocation patterns (**g** two-sided Pearson's chi-squared test; $P_{0-100 \; \mu m} = 3.3 \times 10^{-13}$, $P_{100-250 \; \mu m} = 2.2 \times 10^{-16}$) and the migratory distance toward the meninges (**h** two-sided Steel–Dwass test; $n = 30$, 13, and 20 cells [left to right]; $P_{0-100 \; \mu m} = 0.004$, $P_{100-250 \; \mu m} = 1.5 \times 10^{-6}$). **i** FACS analysis for pallial cells collected from E14 CX3CR1-GFP Tg mice. **j** A schematic showing the coculturing of CX3CR1-GFP[+] microglia with the SVZ/VZ and CP explants. **k** A bright-field image of microglia (white square space) adjacently cocultured with the explants. **l** Representative images of the monitoring of microglial accumulation in the SVZ/VZ (pink) and CP (yellow) explants. **m** The density of microglia in each explant (two-sided Mann–Whitney $U$ test; $n = 4$ individual experiments; $P = 0.029$ for 0.5 day, $P = 0.029$ for 1 day, and $P = 0.029$ for 3 days). Data are presented as the mean values ± S.D. Scale bar, 50 μm. Source data are provided as a Source Data File.

mice, which was previously shown to exhibit normal microglial density on E18.5[28], the relative proportion of microglia was significantly greater on E15 (Fig. 3f, g) (despite a slight decrease in the total number of microglia (Fig. 3h), as reported previously[11]). Also, in the zone immediately below the CP (i.e., bin 3), the relative proportion of microglia was increased, while that of microglia in the inner regions containing the VZ (i.e., bin 1) and the SVZ (i.e., bin 2) was lower than that in the control (Fig. 3f, g). Such abnormalities in microglial distribution on E15 were reproduced in AMD3100-treated WT mice, with no significant decrease in the total number of intrapallial microglia (Supplementary Fig. 6).

To test whether microglial bidirectional migration is perturbed in *Cxcr4*[−/−] mice, live observation was performed in slices from *Cxcr4*[−/−] mice that had been crossed with CX3CR1-GFP mice. The basalward migration of microglia in the outer region

was markedly reduced compared with that in CX3CR1-GFP/*Cxcr4*[+/+] slices, and there were more stationary and apicalward-migrating microglia, while the apicalward migration of microglia in the IZ was strikingly inhibited (Fig. 3i–l; Supplementary Fig. 7; Supplementary Movies 6 and 7). Such abrogation of bidirectional microglial migration was reproduced by the addition of AMD3100 to the culture media of slices from CX3CR1-GFP/*Cxcr4*[+/+] mice (Supplementary Fig. 8). Together, these data indicate that the CXCL12/CXCR4 system plays a pivotal role in the attraction of microglia and their proper positioning in the midembryonic cerebral wall.

**Ectopically added microglia disturbed the CP differentiation.** Although microglia exit the CP in the midembryonic stage, they reenter the CP in the late-embryonic stage (E17). Hence, we

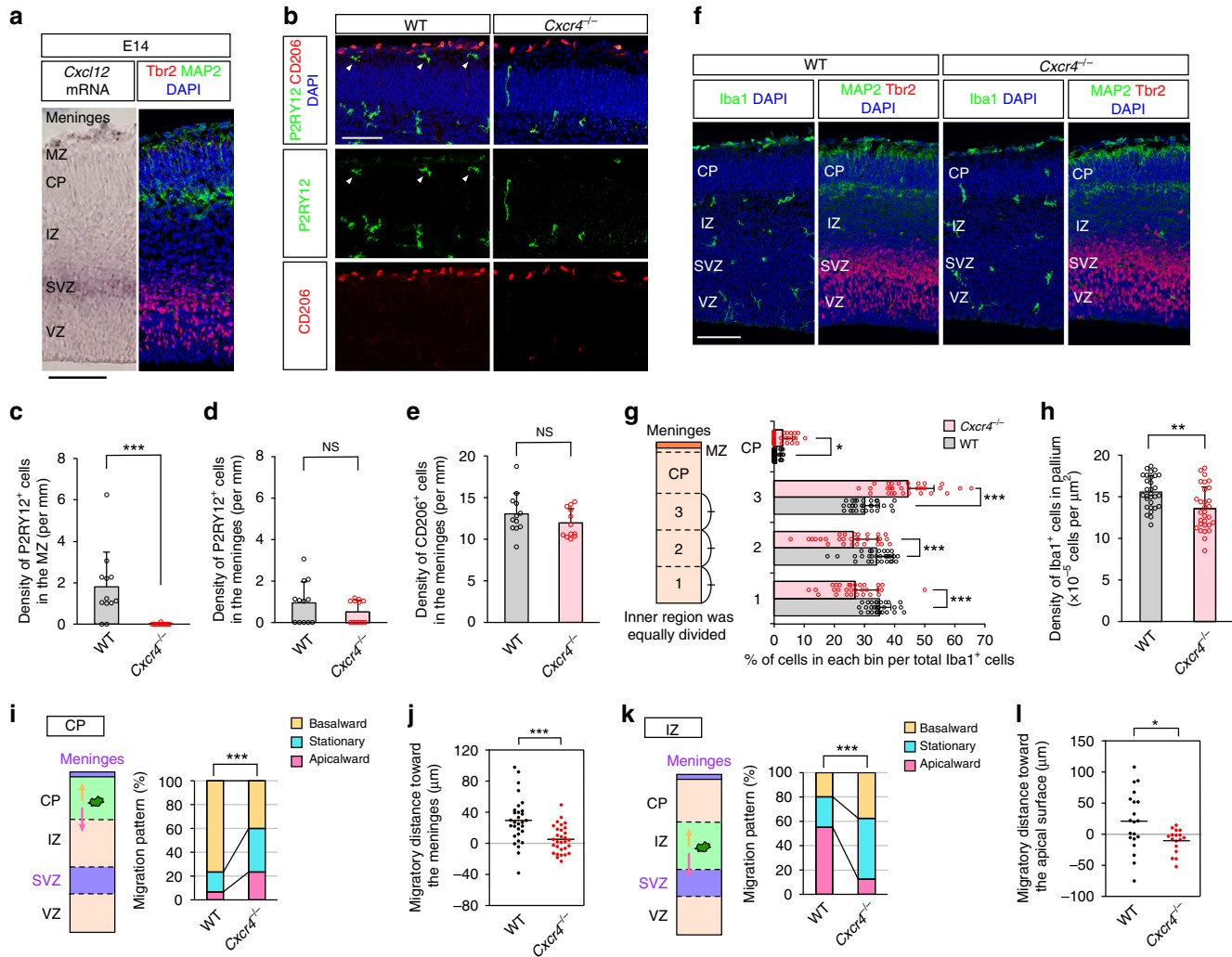

**Fig. 3 The CXCL12/CXCR4 system plays a pivotal role in the bidirectional exit of microglia from the CP and their proper positioning in the cerebral wall. a** In situ hybridization for *Cxcl12* mRNA expression (left) and immunostaining for Tbr2/MAP2/DAPI (right) in the E14 cerebral wall. Scale bar, 100 μm. **b** Immunostaining for P2RY12/CD206/DAPI in E15 cerebral walls. Arrowheads, microglia in the MZ. Scale bar, 100 μm. The density of P2RY12+ microglia in the MZ (**c** $P = 6.7 \times 10^{-5}$) and in the meninges (**d** $P = 0.286$) and of CD206+ macrophages (**e** $P = 0.198$) (two-sided Mann–Whitney $U$ test; $n = 12$ sections from four mice). **f** Immunostaining for Iba1/DAPI (left) and MAP2/Tbr2/DAPI (right) in E15 cerebral walls. Scale bar, 100 μm. The percentage of Iba1+ cells in each bin (the inner part of the cerebral wall than the CP was equally divided into three parts) (**g** $P = 9.2 \times 10^{-10}$ [bin 1], $P = 6.1 \times 10^{-5}$ [bin 2], $P = 3.4 \times 10^{-10}$ [bin 3], $P = 0.045$ [bin CP]) and the density of pallial Iba1+ cells (**h** $P = 0.0019$) (two-sided Mann–Whitney $U$ test; $n = 30$ sections from six mice). Microglial translocation patterns in the CP (**i** two-sided Pearson's chi-squared test; $P = 7.2 \times 10^{-7}$) and the migratory distance toward the meninges (**j** two-sided Mann–Whitney $U$ test; $n = 30$ cells; $P = 1.5 \times 10^{-4}$). Microglial translocation patterns in the IZ (**k** two-sided Pearson's chi-squared test; $P = 1.7 \times 10^{-9}$) and the migratory distance toward the apical surface (**l** two-sided Mann–Whitney $U$ test; $n = 20$ WT cells, 17 *Cxcr4*−/− cells; $P = 0.017$). Data are presented as the mean values ± S.D. Source data are provided as a Source Data File.

sought to understand why they need to transiently exit the midembryonic CP. Neuronal differentiation proceeds in a step-by-step manner through a series of transcriptional waves whose proper sequence is critical for guiding neurons toward their final status[29]. Such a transcriptional wave is considered to occur even at the period that neurons have arrived at the CP as a final wave. Moreover, since neuronal subtype specification occurs not only in progenitor cells but also in postmigratory neurons that form the CP[18–20], we hypothesized that the developing cortex expels microglia that would otherwise disturb neuronal differentiation that occurs in a limited developmental time window. To address this, we first sought to investigate the expression pattern of subtype-associated transcription factors by exposing CP neurons to microglia with two separate slice culture approaches.

First, microglia that were originally present in the inner region of cerebral wall slices (and normally remain there) were attracted

basalward via excess CXCL12 provided by multiple beads, which were densely placed near the meninges (Fig. 4a, b). The live imaging of brain slices derived from E15 CX3CR1-GFP mice demonstrated that CXCL12-soaked beads effectively recruited CX3CR1-GFP+ microglia to the CP within a few hours (Fig. 4c; Supplementary Movie 8). Our direct live monitoring confirmed that many microglia were in the CP for at least several hours. Such attraction by CXCL12 beads was not observed in slices prepared from *Cxcr4*−/− mice (Supplementary Fig. 9). Although most microglia seemed to then return to the IZ/SVZ (probably due to the exhaustion of the CXCL12 in beads), we were able to identify CP regions in which microglia were still present even after 24 h (Fig. 4d, e). Immunohistochemical analyses were performed on these microglia-containing CPs to detect the expression of Ctip2, Satb2, Cux1, and Tbr1 (Fig. 4f). Strikingly, the density of cells expressing Ctip2, a transcription factor that is

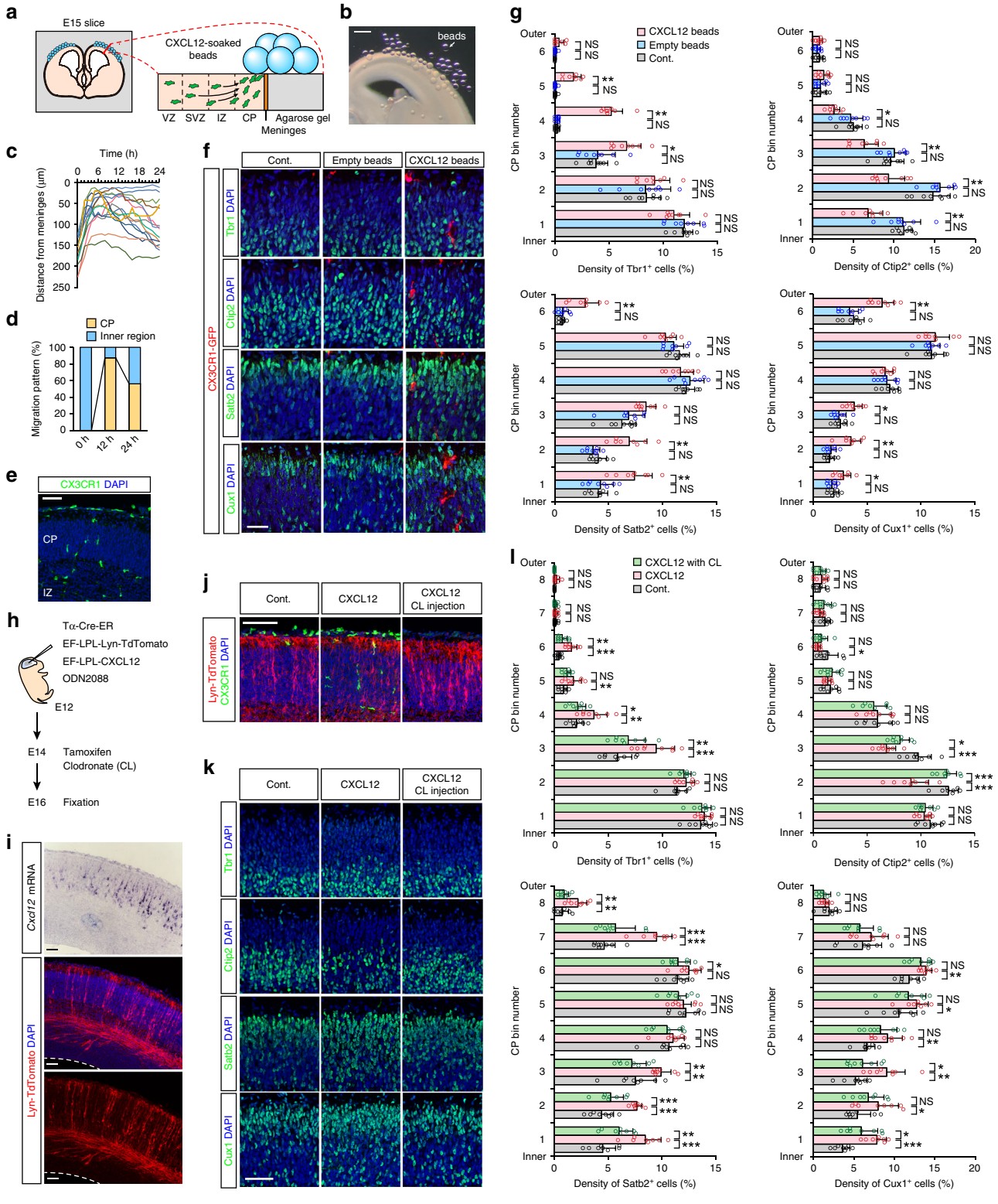

expressed at high levels in L5 neurons in the embryonic stage[16,30], was significantly lower in the deep (bins 1 and 2) to middle (bins 3 and 4) zones of microglia-containing CPs than in CPs from control or empty bead-treated slices (Fig. 4f, g; Supplementary Figs. 10 and 11; Supplementary Table 1). Of note, the density of cells expressing Satb2, which is expressed by a subset of upper-layer (UL) (L2–4) neurons[30,31], was significantly higher in microglia-containing CPs than in microglia-free CPs (both in the deep [1 and 2] and superficial [6] bins). The density of cells

positive for Cux1, which is expressed in UL neurons[32], also increased in both the deep (bins 1–3) and superficial (bin 6) zones of microglia-containing CPs. Moreover, the density of cells expressing Tbr1, which is expressed in L6 and subplate neurons[33], was increased in the upper (bins 3–5) zone in CXCL12 bead-treated slices.

In a second approach, FACS-sorted CX3CR1-GFP+ microglia were transplanted into the CP of isolated ICR mouse brains, and the brains were then sliced for culturing. At 24 h after

**Fig. 4 Artificial delivery of microglia into the CP disturbed the expression of subtype-associated transcription factors in postmigratory neurons. a** A schematic showing the strategy of microglial attraction into the CP in E15 CX3CR1-GFP cortical slices using CXCL12-soaked beads. **b** An image of a cortical slice on which CXCL12-soaked beads were placed. Scale bar, 200 μm. **c** The microglial trajectories (1-h intervals for 24 h) in cultured slices. **d** A graph depicting the proportion of microglia localized in the CP or the inner region (VZ/SVZ/IZ) during culture. **e** Immunostaining for GFP (CX3CR1; green) and DAPI (blue) of the slice 24 h after culturing showing that microglia were attracted to the CP. **f** Representative immunostaining (shown in pseudo color) for GFP (CX3CR1; red), Tbr1/Ctip2/Satb2/Cux1 (green), and DAPI (blue) in cultured slices on which empty or CXCL12 beads were placed and in control slices without beads. **g** The density of cells expressing each marker per total (DAPI$^+$) cells in 20 μm bins in the CP, which was divided into six zones and numbered from the inside. **h** The experimental design for in vivo CXCL12 overexpression in postmigratory neurons. **i** Two adjacent sections of a representative E16 CXCL12-overexpressing brain. In situ hybridization for *Cxc12* mRNA (top) and anti-RFP antibody immunostaining (for Lyn-TdTomato) (lower two panels) are shown. **j** Immunostaining for RFP (Lyn-TdTomato), GFP (CX3CR1), and DAPI. In Cont. group, only Lyn-TdTomato was overexpressed in the brain. In CXCL12 and CXCL12 with CL injection groups, CXCL12 was overexpressed together with Lyn-TdTomato with or without intraventricular injection of CL. **k** Representative immunostaining for Tbr1/Ctip2/Satb2/Cux1 and DAPI in the CP of control and CXCL12-overexpressing brains with or without clodronate administration. **l** The density of cells expressing each marker per total (DAPI$^+$) cells in 20 μm bins in the CP, which was divided into eight zones. For **g** and **l**, two-sided Steel–Dwass test was applied ($n = 8$ sections from four mice; \*\*\**P* < 0.001, \*\**P* < 0.01, \**P* < 0.05, or NS not significant). Data are presented as the mean values ± S.D. The exact *P* values are summarized in Supplementary Table 1. Except for in **b**, the scale bar indicates 50 μm. Source data are provided as Source Data File.

transplantation, the injected microglia remained in the CP, although some had migrated out (Supplementary Fig. 12a–c; Supplementary Movie 9). Immunohistochemical analyses (at 24 h) performed on CPs containing transplanted microglia revealed that the density and the position of cells expressing Ctip2, Satb2, Cux1, and Tbr1 were altered in a manner very similar to those observed in microglia-containing CPs exposed to CXCL12 beads; Tbr1$^+$, Satb2$^+$, or Cux1$^+$ cells were more abundant, whereas Ctip2$^+$ cells were less abundant (Supplementary Fig. 12d–f).

We further performed in vivo experiments to artificially deliver microglia to the CP (Fig. 4h). A vector that allowed the conditional expression of CXCL12 based on Cre-ERT2 under the Tα1-promoter (a neuron-specific promoter) was introduced by in utero electroporation (IUE) on E12, and this delivery was followed by maternal tamoxifen administration on E14. The successful expression of *Cxcl12* mRNA was detected in a pattern matching that of Lyn-TdTomato. (Fig. 4i). On E16, abnormal microglial accumulation was found in CXCL12-overexpressing CPs (Fig. 4j; Supplementary Fig. 13). Immunohistochemical analyses demonstrated that Satb2$^+$ and Cux1$^+$ cells were significantly more abundant in microglia-containing CPs in the deep zone (bins 1–3) than in the controls (Fig. 4k, l; Supplementary Table 1). In addition, the density of Tbr1$^+$ cells was increased in the middle zone (bins 3–5) in CXCL12-overexpressing CPs. In contrast, the density of Ctip2$^+$ cells was significantly lower in the deep zones (bins 2 and 3) of microglia-containing CPs than in the controls. To exclude the possibility that CXCL12 overexpression directly affects neuronal marker expression, we combined this overexpression system with the intraventricular injection of clodronate (which selectively depletes microglia) on E14, and found that the expression of these transcription factors was comparable between the only Lyn-TdTomato-expressing control brains and microglia-depleted CXCL12-overexpressing brains. These results obtained by slice culture-based and in vivo experiments suggest that microglia may influence postmigratory cortical neurons in the expression of neuronal subtype-associated transcription factors, if these cells are inadvertently positioned in the CP, although it is also possible that ectopic (intra-CP) microglia affected neuronal migration and/or positioning.

**In vitro influences of microglia on CP-like neurons**. We next sought to separately examine the influence of microglia on post-migratory neurons using a different coculture system. Instead of exposing bona fide (in situ) CP neurons to microglia (as in the experiments above), we used Gadd45g-d4Venus mice to prepare neurons corresponding to CP neurons in vitro based on the disappearance of d4Venus fluorescence during culture (within

2 days)[34] (Fig. 5a–c) and then thoroughly (much more extensively than in slice-based cocultures) mixed these d4Venus$^-$ (chronologically CP-like [see "methods"]) neurons with CX3CR1-GFP$^+$ microglia (Fig. 5d; Supplementary Fig. 14). CP-like neurons cocultured with or without microglia for 24 h were then analyzed immunocytochemically. The proportion of βIII-tubulin$^+$ neurons that expressed Ctip2 was significantly smaller in microglia-containing cultures (44.5%) than in control (microglia-free) cultures (56.3%) (Fig. 5e, f). In contrast, the proportion of βIII-tubulin$^+$ neurons that expressed Tbr1, Satb2, or Cux1 was higher in microglia-containing cultures (40.8%, 34.5%, and 24.4%, respectively) than in control cultures (29.5%, 22.8%, and 12.7%, respectively). Importantly, the in vitro microglial exposure of neurons that were younger (0 day, Gadd45g-d4Venus$^+$) (correspond to neurons in the SVZ/IZ in vivo[31]) than CP-like neurons (2 days) did not disturb the expression of Tbr1, Ctip2, Satb2, or Cux1 (Fig. 5g–i), suggesting that cortical neurons may be more sensitive to microglial influence after their arrival in the CP rather than during migration.

Since we only examined the single staining of specific four neuronal markers (Tbr1, Ctip2, Satb2, and Cux1) through the immunohistochemical (Fig. 4) and immunocytochemical (Fig. 5) analyses, we next performed FACS analysis for the detection of multiple different markers in individual neurons. Gadd45g-d4Venus$^-$ CP-like neurons were cultured in the presence or absence of microglia collected based on their surface expression of CD11b (ITGAM) using a magnetic separation system (MACS) (Fig. 6a). At 24 h, cells that were negative for CD11b were isolated to remove microglia from the coculture samples, and were assessed for the intracellular staining for FACS.

In the neurons cocultured with microglia (neurons$^{MG}$) group, the frequency of cells that were Brn2$^+$Satb2$^+$ or Satb2$^+$RORβ$^+$ was significantly increased, and that of cells triple-positive for Brn2, Satb2, and RORβ was increased compared with those in the neurons cultured alone (neurons$^{Cont}$) group (Fig. 6b–d; Supplementary Fig. 15). The frequency of cells that were double-positive for Cux1 and Tbr1 was greater in the neurons$^{MG}$ group than in the neurons$^{Cont}$ group. In contrast, the number of cells that were Ctip2$^+$ but negative for Satb2 was slightly but significantly decreased in the neurons$^{MG}$ group. Taken together, both of our immunostaining and FACS analyses strongly suggest that postmigratory neurons aberrantly increased the expression of UL neuron markers.

**RNA sequencing (RNA-Seq) revealed wider microglial influences on CP neurons**. To evaluate more broadly changes in gene expression for fully understanding of microglial influence on differentiation of postmigratory neurons, we sought to perform

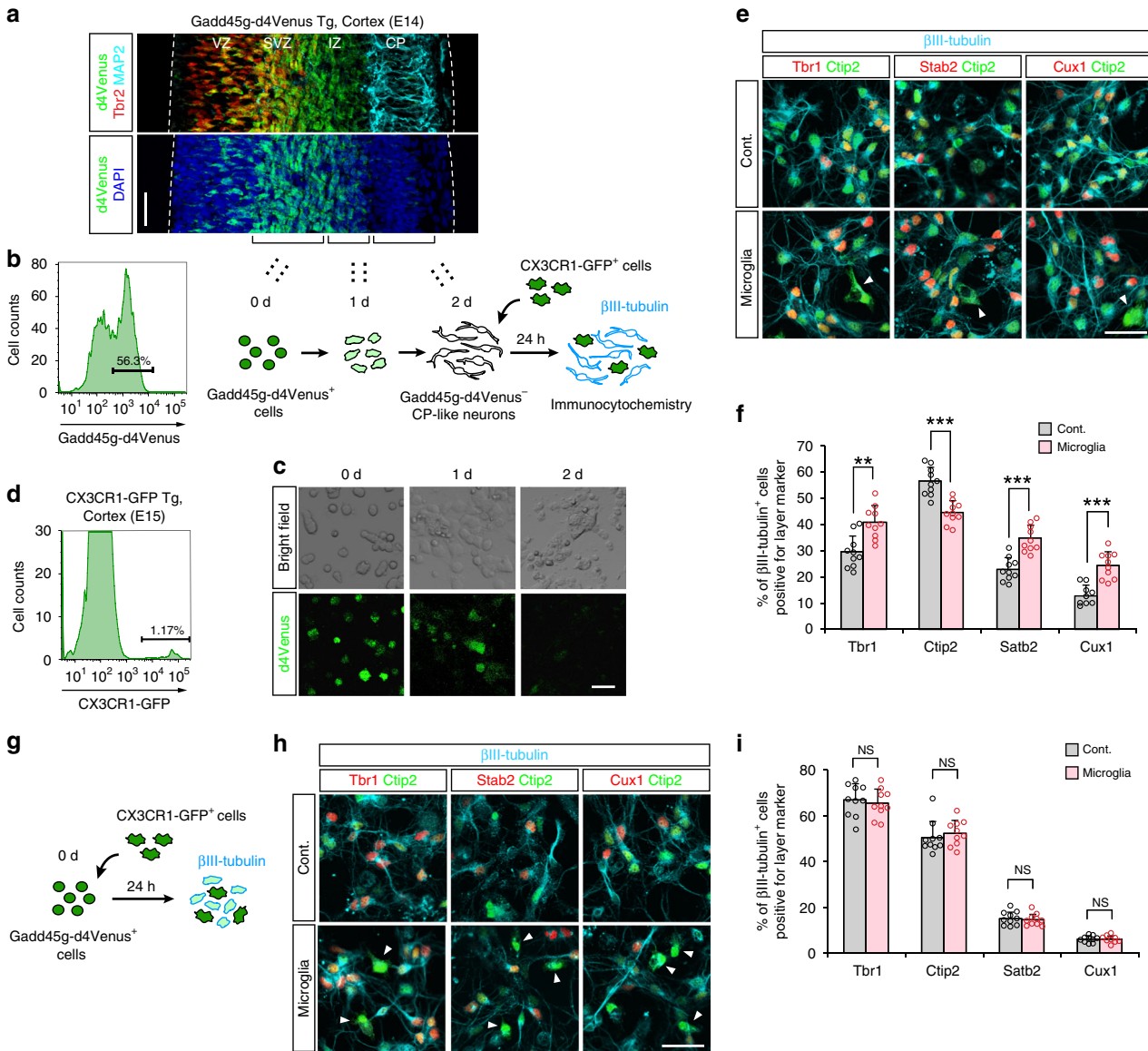

**Fig. 5 In vitro-prepared CP-like neurons changed subtype marker expression upon exposure to microglia. a** Immunostaining for d4Venus (Gadd45g; green), Tbr2 (red), MAP2 (cyan), and DAPI (blue) in E14 Gadd45g-d4Venus Tg mouse cerebral walls. The broken lines indicate apical and basal surfaces. Scale bar, 50 μm. **b** The protocol for the in vitro preparation of CP-like neurons, which were obtained from Gadd45g-d4Venus⁺ cells isolated from the E14 Gadd45g-d4Venus cortex by FACS, after culture for 2 days. **c** A time-lapse series showing the reduction in Gadd45g-dVenus expression during culture. Scale bar, 20 μm. **d** CX3CR1-GFP⁺ microglia harvested from E15 CX3CR1-GFP pallial walls by FACS were added to the CP-like neuronal cultures. **e** Representative immunostaining (shown in pseudo color) for βIII-tubulin (cyan), Tbr1/Satb2/Cux1 (red), and Ctip2 (green). The arrowhead indicates microglia. Scale bar, 30 μm. **f** The proportion of the βIII-tubulin⁺ cells that expressed Tbr1, Ctip2, Satb2, or Cux1 in cocultures with or without microglia (two-sided Mann–Whitney U test; n = 10 independent experiments; P = 0.0021 for Tbr1, P = 7.6 × 10⁻⁵ for Ctip2, P = 7.6 × 10⁻⁵ for Satb2, P = 1.5 × 10⁻⁴ for Cux1). **g** An experimental schematic of the coculture of youngest (0 day, Gadd45g-d4Venus⁺) neurons and CX3CR1-GFP⁺ microglia. **h** Representative immunostaining (shown in pseudo color) for βIII-tubulin (cyan), Tbr1/Satb2/Cux1 (red), and Ctip2 (green). The arrowhead indicates microglia. Scale bar, 30 μm. **i** The proportion of βIII-tubulin⁺ cells that expressed Tbr1, Ctip2, Satb2, or Cux1 in cocultures with or without microglia (two-sided Mann–Whitney U test; n = 10 independent cultures; P = 0.529 for Tbr1, P = 0.289 for Ctip2, P = 0.631 for Satb2, P = 0.912 for Cux1). Data are presented as the mean values ± S.D. Source data are provided as a Source Data File.

RNA-Seq analysis using the abovementioned high-throughput coculture system, which is considered to enable us to efficiently obtain more neurons that were homogeneously exposed to microglia than the samples prepared from in vivo and slice culture-based experiments.

As same as in the case of preparation for FACS analysis (Fig. 6a), Gadd45g-d4Venus⁻ CP-like neurons were cultured with/without isolated CD11b⁺ microglia for 24 h. After CD11b⁺ cells were removed from the coculture samples, total RNA was extracted from cultured neurons. The expression level of *Itgam* (*Cd11b*) was extremely low in the neurons[MG] group (Fig. 7a), demonstrating the successful depletion of CD11b⁺ microglia from the cocultures before the collection of samples for RNA. Genes commonly expressed in macrophages and microglia (*Cx3cr1, Aif1 (Iba1), Csf1r*, etc.) or microglia-specific genes (*P2ry12, Tmem110, Fcrls*, etc.)[35,36] were not detected or were expressed at very low levels, and the expression of these genes was comparable between the neurons[Cont] and neurons[MG] groups.

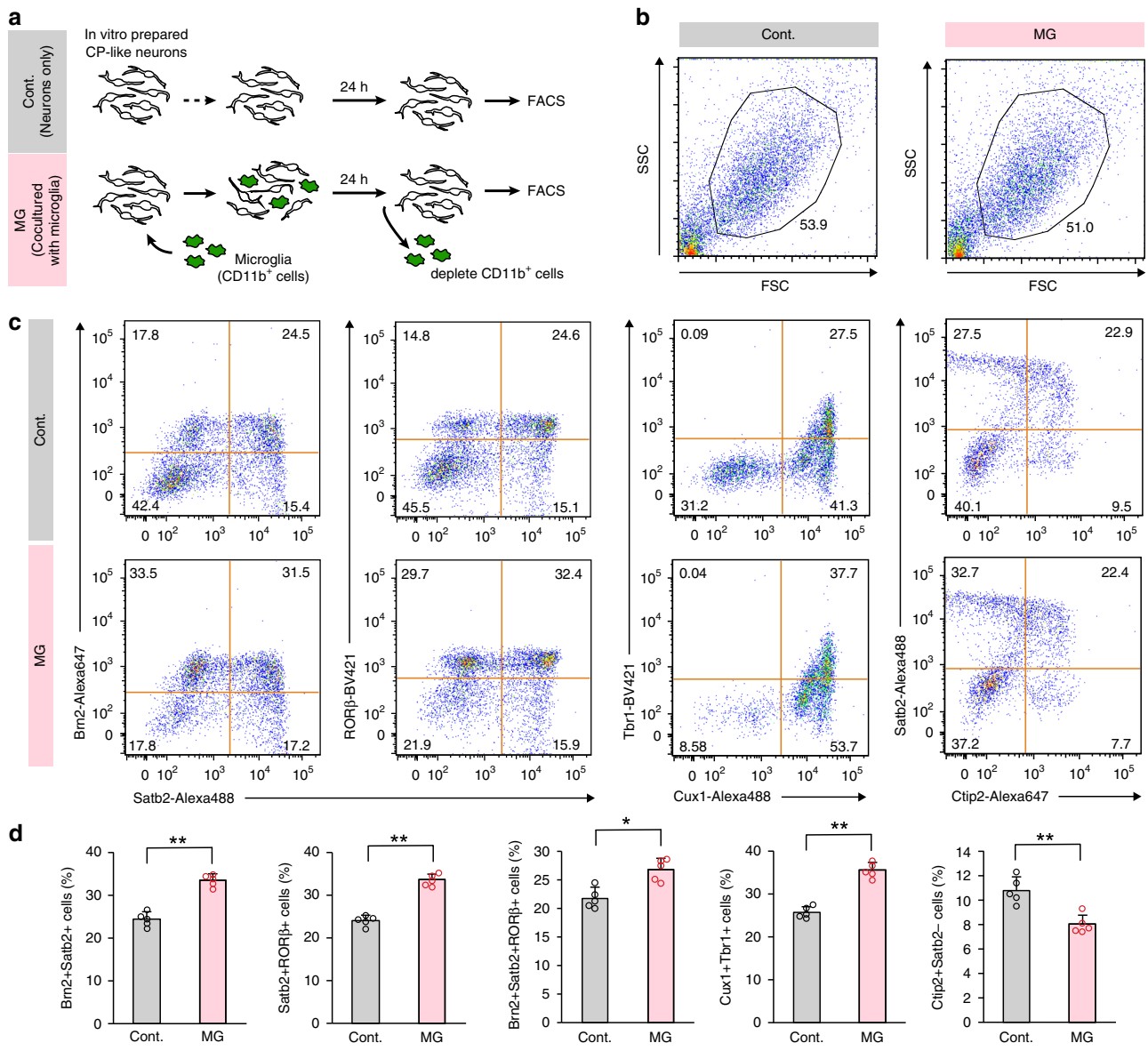

**Fig. 6 FACS analysis for the detection of multiple subtype-associated transcription factors in microglia-cocultured neurons. a** A schematic depicting the cell preparation technique used for FACS analyses. Neurons[MG], cultures of CP-like neurons that were cocultured with microglia; neurons[Cont], cultures without microglia. **b** FSC/SSC plots of neurons cultured with/without microglia, which were treated for intracellular staining. **c** Representative data of FACS analysis of in vitro-prepared CP-like neurons after coculture with or without microglia. **d** Graphs showing the average percentage of Brn2+Satb2+, Satb2+RORβ+, Brn2+Satb2+RORβ+, Cux1+Tbr1+, and Ctip2+Satb2− cells, as determined by FACS (two-sided Mann–Whitney $U$ test; $n = 5$ independent cell cultures; $P = 7.9 \times 10^{-4}$ for Brn2+Satb2+ cells, $P = 7.9 \times 10^{-4}$ for Satb2+RORβ+ cells, $P = 0.016$ for Brn2+Satb2+RORβ+ cells, $P = 7.9 \times 10^{-4}$ for Cux1+Tbr1+ cells, and $P = 7.9 \times 10^{-4}$ for Ctip2+Satb2− cells). Data are presented as the mean values ± S.D. Source data are provided as a Source Data File.

These data validate that our RNA was from purified CP-like neurons with no or negligible microglial contamination. Then, we compared 46 principal genes, which are uniquely expressed in particular cortical layer(s) and important for neuronal differentiation in the embryonic stage.

Heat map data revealed that most of UL marker genes (*Satb2, Mdga1, Frmd4b, Cux2, Btg1,* and *Inhba*) were more strongly expressed in neurons[MG] than in neurons[Cont] (Fig. 7b; Supplementary Fig. 16; Supplementary Table 2), although there were some exceptions; the expression of *Plxnd1* and *Lpl*, which are known as L2/3 and L5 marker, was decreased. Meanwhile, the expression of many of DL marker genes (*Tle4, Lmo3, Foxo1, Nr4a3, Grb14, Sox5, Sla, Lxn, Cdh13,* and *Pcp4*) was decreased. These data showed a tendency to reduce the expression of DL

marker genes and increase the expression of typical UL marker genes. In addition, the expression of *Tbr1* was increased in neurons[MG] compared with that in neurons[Cont], which is consistent with the results obtained from the immunohistochemical analyses.

The aforementioned immunocytochemical data showed that the proportion of Ctip2+ in neurons[MG] was decreased, but our RNA-Seq analysis revealed that the expression of *Bcl11b (Ctip2)* was conversely increased. In addition, the expression of *Cux1*, whose expression was increased in immunocytochemical analysis, in neurons[MG] was comparable with neurons[Cont] in RNA-Seq analysis. These inconsistencies would reflect that bulk RNA-Seq, even performed for neurons enriched based on culture from the same SVZ region, may have included a large variety of cell types,

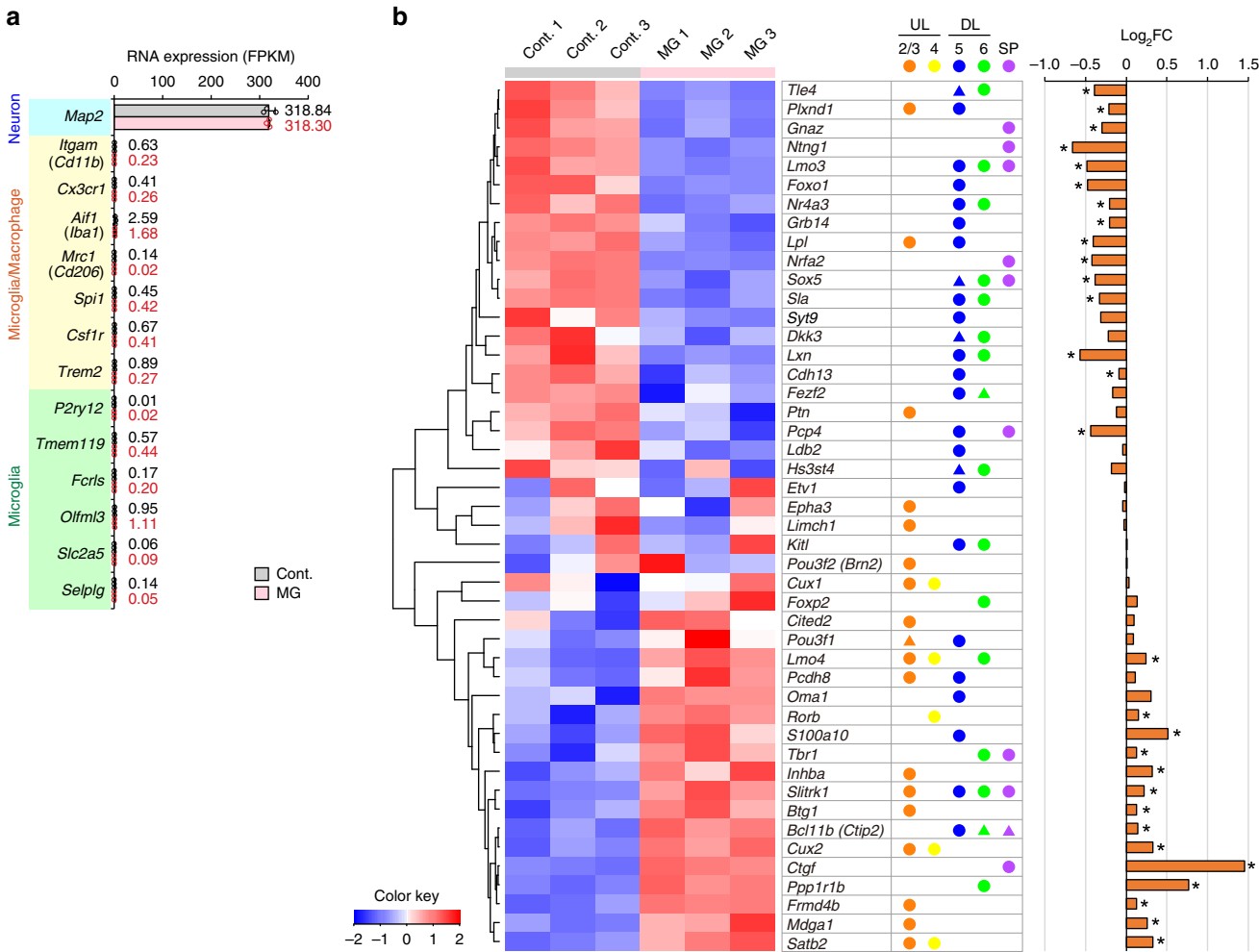

**Fig. 7 Microglia widely disturbed the expression of transcription factors and molecules essential for the functional maturation of cortical neurons. a** A comparison of the FPKM values of the neuron-specific marker gene (MAP2) and several genes known as common microglial and macrophage markers (shown in yellow) and microglia-specific markers (green) between neurons[Cont] and neurons[MG]. The black and red numbers beside the bar graph show the FPKM values for each gene in neurons[Cont] and neurons[MG], respectively ($n = 3$ samples obtained from independent cell cultures). Data are presented as the mean values ± S.D. **b** A heat map of 46 principal genes, which are uniquely expressed in particular cortical layer(s) and important for neuronal differentiation, showing the neurons[MG]/neurons[Cont] expression ratio. The middle panel shows the cortical layer identity (L2/3, L4, L5, L6, and SP [subplate]) of each gene. Circle shows that the gene is known to be strongly expressed in the corresponding layer; triangle shows its weak expression (see Supplementary Table 2). The right panel indicates the log$_2$ fold change (FC) of gene expression (neurons[MG]/neurons[Cont]). FDR < 0.1 is considered significant (*). Source data and the exact FDR values are provided as a Source Data File and Supplementary Data 1.

where as immunocytochemistry had a single-cell resolution. Or, they might be explained by the possible effects of microglia on the posttranscriptional regulation of neurons.

To comprehensively investigate changes in the expression of genes important for neuronal differentiation, we compared the transcriptomic profiles of the neurons[Cont] and neurons[MG] groups based on differentially expressed gene (DEG) analysis. Among the genes that were reliably detected in both groups (11,603 genes), we identified 143 downregulated genes and 246 upregulated genes in neurons[MG] group based on the statistical threshold (log$_2$ fold change [FC] > 1.4, FDR < 0.1) (Fig. 8a). Gene ontology (GO) analysis for significantly downregulated genes showed that those were categorized as "anterograde transsynaptic signaling", "cell–cell signaling", "transport", and "regulation of localization" (Fig. 8b; Supplementary Table 3). Of note, many of genes involved in the GABAergic synapse transmission, especially those which functionally work in postsynaptic pyramidal and non-pyramidal cells (*Gabra3*, *Gabra5*, *Gabrb1*, *Gabrd*, *Kcnj6*, *Trak2* etc.), were markedly reduced. Our culture system used

for RNA-Seq analysis may have contained not only the pallium-derived (presumptive excitatory) neurons but also interneurons derived from the ganglionic eminences (GEs), which was suggested by detection of the expression of *Lhx6* in our RNA-Seq data. This is explained by that GE-derived interneurons migrate tangentially (dorsally) through the pallial SVZ[37], which was entirely labeled with Gadd45g-d4Venus. Our RNA-Seq data further showed that somatostatin (*Stt*) and calretinin (*Calb2*), molecules characterizing GABAergic inter-neuron subtypes, were significantly decreased in the neurons[MG] group. In addition, the reductions in the mRNA levels of molecules that participate in the functional maturation of neurons, such as calmodulin 1 (*Calm1*), calmodulin 2 (*Calm2*), *Gap43*, cytochrome oxidase subunit 6B1 (*Cox6b1*), and synapto-physin (*Syp*), were also detected in neurons[MG] group. On the other hand, we found upregulation of mRNA levels of molecules necessary for morphological maturation, such as tubulins (multiple classes) and collapsin response mediator protein 1 (*Crimp1*), in the neurons[MG]. Despite such an apparently

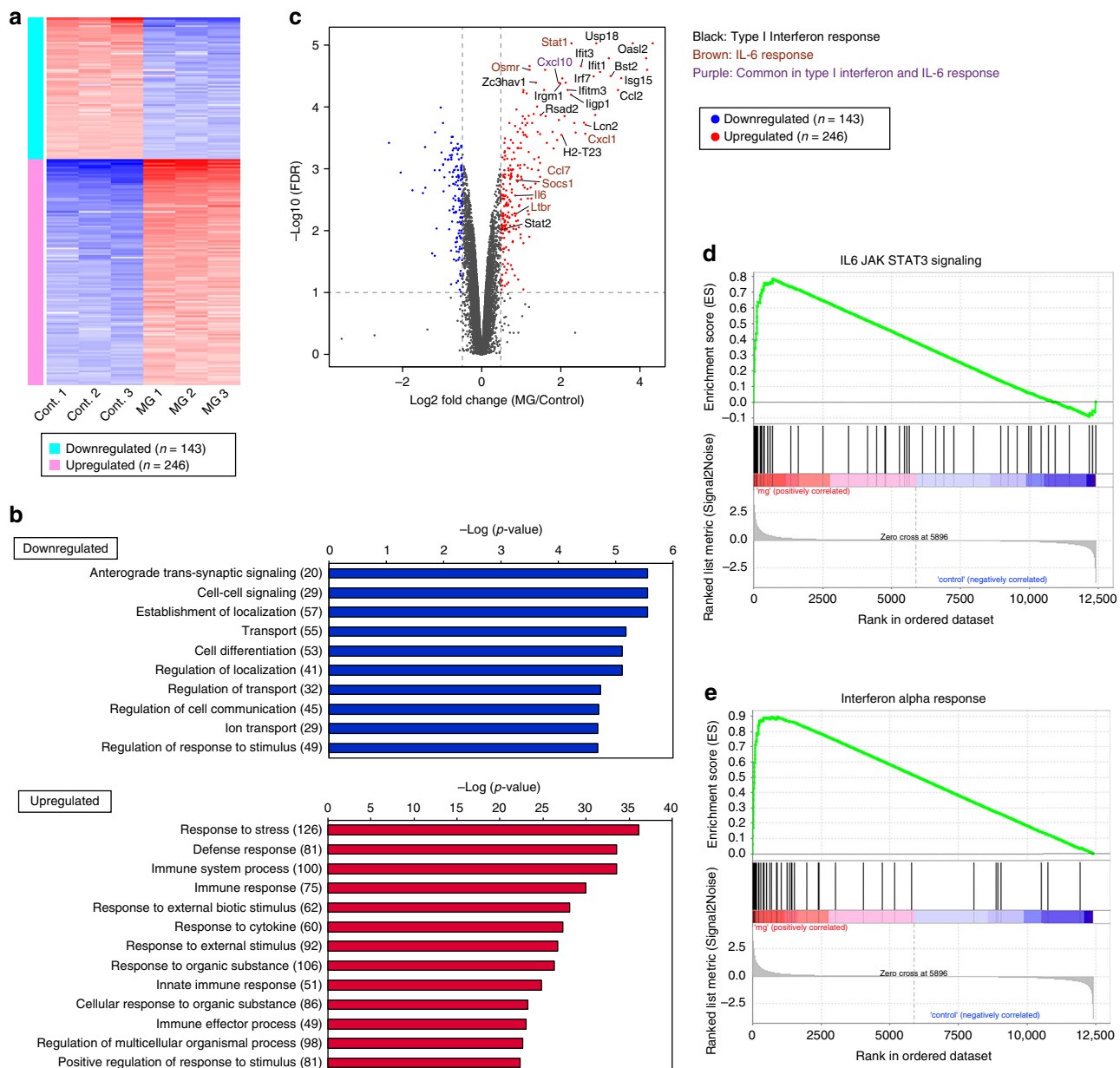

**Fig. 8 Signaling pathways triggered by IL-6 and IFN-I were activated in CP-like neurons after coculture with microglia. a** Transcriptomic profiles of the neurons[Cont] and neurons[MG] groups based on differentially expressed gene (DEG) analysis. Among the genes that were reliably detected in both groups (11,603 genes), we identified 143 downregulated genes and 246 upregulated genes in neurons[MG] group based on the statistical threshold (log$_2$ fold change [FC] > 1.4, FDR < 0.1). **b** Gene ontology (GO) analysis for significantly downregulated and upregulated genes (see Supplementary Table 3). **c** A volcano plot depicting DEGs between the neurons[Cont] and neurons[MG] groups. The negative log of FDR is plotted on the Y-axis, and the log$_2$ FC of neurons[MG]/ neurons[Cont] is plotted on the X-axis. The blue and red points on this graph represent mRNAs that were significantly downregulated ($n = 143$) and upregulated ($n = 246$), respectively, in neurons[MG] (FC > 1.4; FDR < 0.1). The color of the gene names indicates that the genes are typical in the type I interferon response (black), typical in the IL-6 response (brown), and common in type I interferon and IL-6 responses (purple). Gene set enrichment analysis showing a barcode plot for genes registered in IL-6 JAK STAT3 signaling (**d**) or the interferon alpha response (**e**). Genes were ranked based on their differential expression between neurons[MG] and neurons[Cont]. A plot of running enrichment score (RES) is shown in green (top). The vertical black bars (middle) indicate a gene within each gene set. The correlation of the expression pattern of genes belonging to each subcluster is shown in the gray histogram (bottom). The RNA-Seq data are summarized in Supplementary Data 1.

positive influence of aberrant microglia on a cell-structural aspect of cortical neurons, our data together demonstrated that abnormal exposure to microglia widely disturbed the expression of subtype-associated transcription factors and reduced the expression of molecules involved in the functional maturation of cortical neurons.

**CP neurons were sensitive to microglia-derived IL-6 and IFN-I.** To search for the molecular mechanisms underlying the disturbed expression pattern of neuronal subtype-associated genes in the presence of microglia, we focused on the upregulated genes in the neurons[MG] group in the DEG analysis. GO analysis showed that families of genes that encode proteins characteristic of the

immune response (categorized as "immune system process", "response to cytokine", and "innate immune response") were highly induced in neurons[MG] (Fig. 8a, b; Supplementary Table 3). Notably, the expression of genes relevant to the type I interferon (IFN-I) and IL-6 response were upregulated in neurons[MG] (Fig. 8c). Gene set enrichment analysis (GSEA) of the DEGs demonstrated that the gene sets categorized as "IL-6 JAK/STAT3" and "interferon alpha response" were enriched in the upregulated genes in neurons[MG] (Fig. 8d, e), suggesting that IL-6 and IFN-I are the most likely candidates for modulators of neuronal subtype-associated gene expression.

We next asked whether these cytokines are secreted by microglia cocultured with neurons. IL-6 transduces signals via functional receptor complexes composed of the IL-6 receptor that take a membrane-bound form (IL6Ra) or a soluble form (sIL6) and the IL-6 signal transducer (IL6ST [gp130])[38]. IFN-I, which has about 20 subtypes (i.e., 14 IFN-α, single IFN-β, and several poorly defined single subsets in the mouse[39]), signals through a heterodimeric transmembrane receptor composed of interferon α/β receptor subunit (IFNAR) 1 and IFNAR2. We quantified IL-6, IFN-α2/α4, and IFN-β levels in the supernatant of the cocultures of CP-like neurons and microglia by enzyme-linked immunosorbent assay (ELISA). IL-6 and IFN-β were detected at concentrations of 52.1 and 34.8 pg ml$^{-1}$, respectively, whereas IFN-α2/α4 was below the detection limit (Fig. 9a). The in situ hybridization database showed that E14 postmigratory neurons in the CP substantially express Il6ra, gp130, and Ifnar1 in the cerebral wall (Fig. 9b). Moreover, Il6ra has been shown to be highly expressed in the deep layer of the adult rat brain[40].

To address whether IL-6 and/or IFN-I participate in the effect of microglia on neuronal subtype-associated gene expression of CP-like neurons, neutralizing/blocking antibodies targeting IL-6 or IFNAR1 were added to neuronal cultures (Fig. 9c). FACS analyses demonstrated that both anti-IL-6- and anti-IFNAR1-neutralizing antibodies almost abolished the alterations in the expression of subtype markers (i.e., decreased the proportion of Satb2+RORβ+, Brn2+Satb2+RORβ+, and Cux1+Tbr1+ cells and increased that of Ctip2+Satb2− cells compared with those in antibody-free neurons[MG]) (Fig. 9d, e; Supplementary Figs. 17 and 18). Exceptionally, the microglia-induced increase in the density of Brn2+Satb2+ cells was not abolished for unknown reasons. Furthermore, these neutralizing antibodies were administered in vivo (intraventricularly) into hemispheres in which CPs had been subjected to the IUE-based overexpression of CXCL12 (to attract microglia) (Fig. 10a). Similarly, the increase in the proportion of Satb2+ or Cux1+ cells that was observed in CPs that were artificially induced to contain microglia in mock-treated mice was significantly attenuated in the CPs (especially in the deep zone [bins 1–3]) of mice treated with anti-IL-6 and/or anti-IFNAR1 antibodies (Fig. 10b, c). In addition, the increase in the density of Tbr1+ cells was almost abrogated in the middle zone (bins 3 and 4) zone. In contrast, the decrease in the density of Ctip2+ cells was recovered in the deep zones (bins 2 and 3) of antibody-treated brains. These data strongly suggest that the disturbed balance of neuronal subtype-associated gene expression caused by ectopic microglia are attributed to IL-6 and IFN-I secreted from microglia.

## Discussion
Here we describe the biological significance of the transient (E15–E16) absence of microglia from the CP, a zone in which postmigratory neurons accumulate. We determined that a key mechanism of this transient exit of microglia from the CP is the CXCL12/CXCR4 system. Through the experiments to artificially expose in vivo CP neurons or in vitro grown CP-like

(age-identified) neurons to excessive microglia, we found that CP neurons failed to appropriately express subtype-associated transcription factors, showing a tendency to reduce the expression of DL marker genes and increase the expression of typical UL marker genes. Since several markers were exceptionally fluctuated against this tendency, we cannot simply conclude that microglia induce CP neurons to fully acquire the properties characteristic in UL neurons, but they widely disturbed the expression pattern of neuronal subtype-associated genes essential for neuronal proper differentiation. Our RNA-Seq data also reveal that CP neurons abnormally exposed to microglia downregulated the expression of molecules involved in neuronal differentiation, such as those important for functional maturation of cortical neurons and synapse transmission, in spite of their positive influence on cell-structural and morphological aspect of cortical neurons. Further, we determined two important mediators, microglia-derived IL-6 and IFN-I, participate in the disturbance of gene expression of neuronal subtype-associated genes in postmigratory neurons (Supplementary Fig. 19).

A previous in vivo study showed that maternal immune activation (MIA) with lipopolysaccharide increases the total number of microglia in the cerebral wall. In that MIA model, however, the number of microglia in the CP did not increase[14] (i.e., microglial absence was still maintained), suggesting that the mechanisms underlying the physiological disappearance of microglia from the CP, as the present study revealed, may be resistant to generally used MIA methods. As such, the experimental abrogation of microglial absence is difficult, and the potential negative/pathological effect of the failure of microglial disappearance from a certain region was previously not known. Thus, we sought to understand this unsolved problem based on our hypothesis that the developing cortex transiently expels microglia that would otherwise disturb neuronal differentiation processes.

We demonstrated that the transient disappearance of microglia from the midembryonic CP is primarily due to the bidirectional attraction of microglia by CXCL12, but two questions remain to be addressed. First, we do not yet know how microglia reenter the CP around E17, despite the fact that the expression of CXCL12 expression in the meninges is sustained through this stage (although CXCL12 expression in the SVZ peaks on E14 and then declines). Since CXCR4 expression in microglia continues until the perinatal period[28], the intracellular processing of CXCL12-evoked signaling required for migration might change and/or be silenced toward the end of the embryonic period. Second, we also observed that some microglia did exit the CP of the cerebral walls of E14/15 Cxcr4$^{-/-}$ mice, which lack the CXCL12/CXCR4 attraction system, implying the existence of other mechanisms. For instance, the chemorepulsion molecule Slit1, which is highly expressed in the embryonic CP[41], could act on microglia via PlexinA1[42]. Hence, the localization of microglia in the cortical wall is spatiotemporally regulated via the complicated mechanisms involving multiple factors, including CXCL12/CXCR4 system, throughout the developmental stage. We discovered though unique approaches that the midembryonic cerebral wall induces microglia to exit the CP, thereby providing a "sanctuary" for postmigratory neurons to ensure their appropriate differentiative processes.

Our molecular characterization coupled with the artificial/experimental exposure of CP neurons to microglia revealed that disturbances in the subtype-associated gene expression were primarily mediated by IL-6 and IFN-I. However, the detailed mechanism by which these cytokines modulate the expression of transcription factors fundamental for neuronal proper differentiation remains unresolved. Of note, mounting evidence has reported the existence of multiple regulation mechanisms via cytokine signaling[43–47]. IL-6 activates signal transducer and activator of

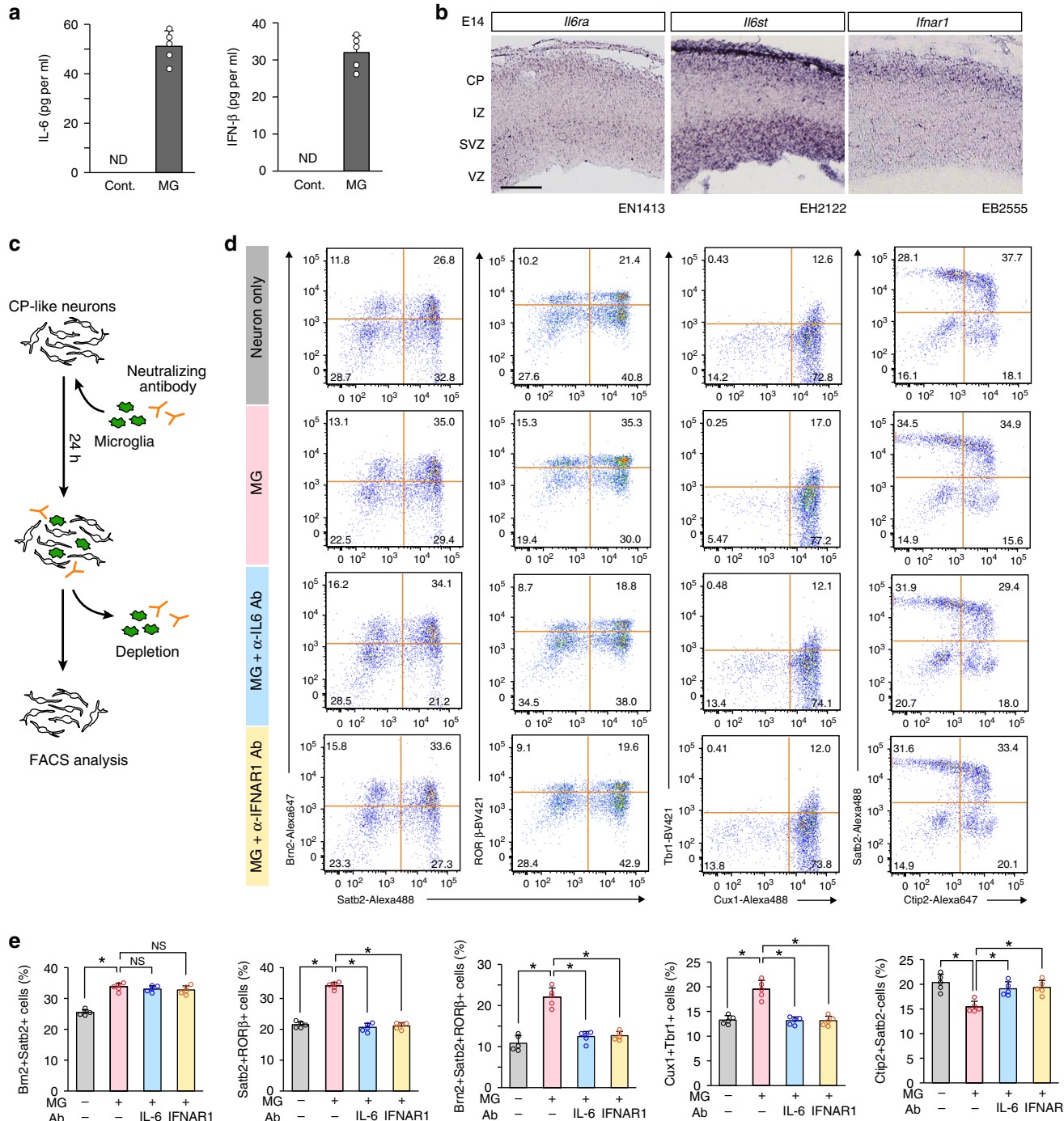

**Fig. 9 Differentiative marker expression in CP-like neurons is sensitive to IL-6 and IFN-I secreted by microglia. a** The quantification of IL-6 and IFN-β released into the culture media of in vitro-prepared CP-like neurons with or without microglia (n = 5 independent cultures; ND not detected). **b** GenePaint mouse atlas of *Il6ra*, *Il6st (gp130)*, and *Ifnar1* mRNA expression in E14.5 mice (https://gp3.mpg.de). GenePaint set IDs are indicated at the bottom right of the panels. Scale bar, 100 μm. **c** An experimental schematic for the IL-6 and IFN-I neutralization assay. In advance, microglia isolated by CD11b magnetic separation were pretreated with anti-IL-6 or anti-IFNAR1 antibodies for 30 min. After coculturing with microglia in the presence of neutralizing antibodies for 24 h, in vitro-prepared CP-like neurons were purified by selecting cells negative for CD11b. **d** Representative FACS analysis data of CP-like neurons after coculture with or without microglia. **e** Graphs depicting the average percentage of Brn2⁺Satb2⁺, Satb2⁺RORβ⁺, Brn2⁺Satb2⁺RORβ⁺, Cux1⁺Tbr1⁺, and Ctip2⁺Satb2⁻ cells (two-sided Steel–Dwass test; n = 5 independent cultures; P = 0.045, 0.836, and 0.721 for Brn2⁺Satb2⁺ cells, P = 0.045, 0.045, and 0.045 for Satb2⁺RORβ⁺ cells, P = 0.045, 0.045, and 0.045 for Brn2⁺Satb2⁺RORβ⁺ cells, P = 0.045, 0.045, and 0.045 for Cux1⁺Tbr1⁺ cells, and P = 0.045, 0.045, and 0.045 for Ctip2⁺Satb2⁻ cells [in the order of control vs microglia-added samples without antibodies, microglia-added samples without antibodies vs those treated with anti-IL-6 antibodies, and microglia-added samples without antibodies vs those treated with anti-IFNAR1 antibodies]). Data are presented as the mean values ± S.D. Source data are provided as a Source Data File.

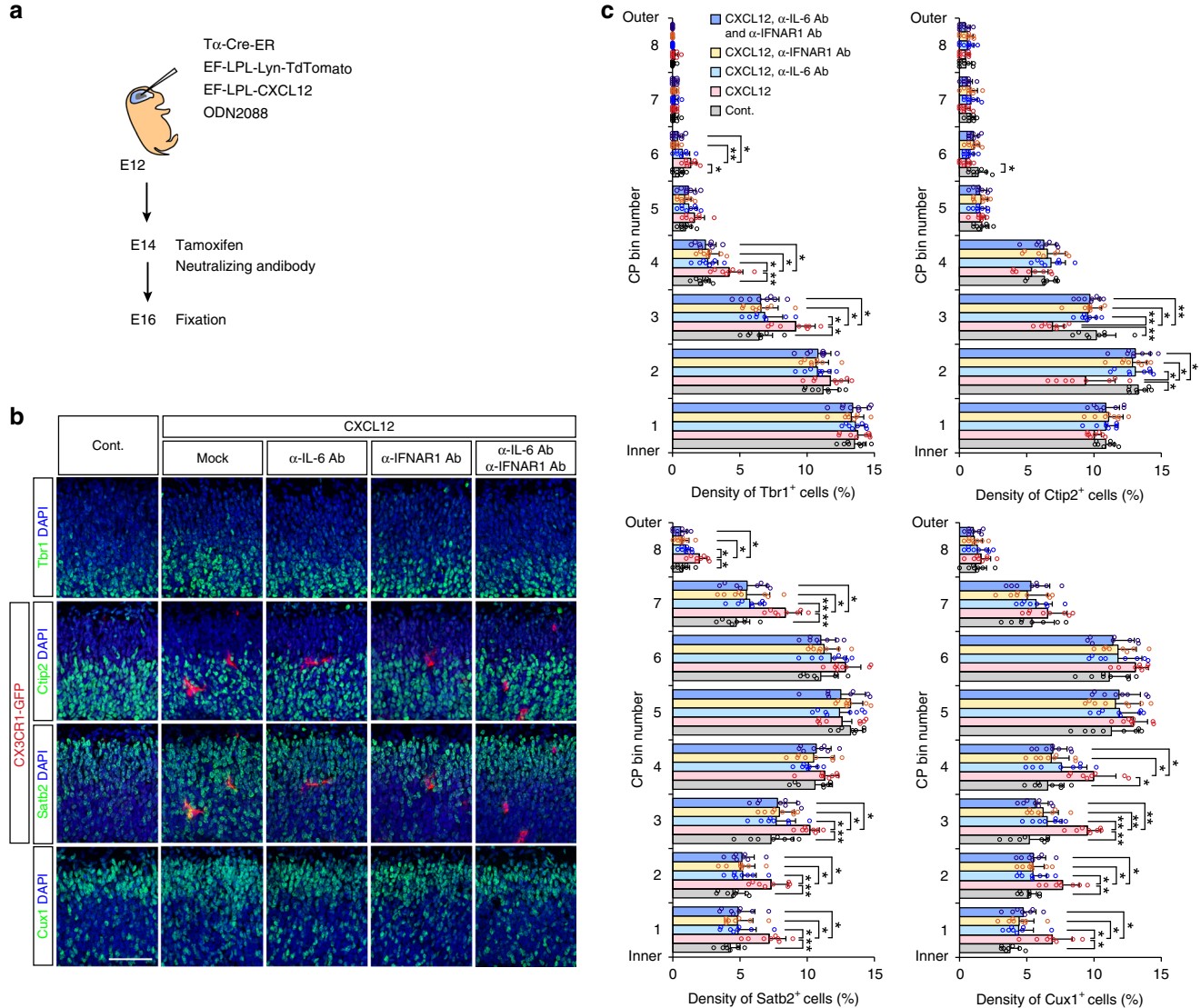

**Fig. 10 In vivo administration of anti-IL-6 and/or IFNAR1 antibodies significantly attenuated the disturbed expression of neuronal subtype-associated transcription factors in postmigratory neurons. a** The experimental design for the administration of neutralizing antibodies, which is combined with in vivo CXCL12 overexpression in postmigratory neurons. **b** Representative immunostaining (shown in pseudo color) for GFP (CX3CR1; red), Tbr1/Ctip2/ Satb2/Cux1 (green), and DAPI (blue) in control and CXCL12-overexpressing brains that were mock-treated or administered neutralizing antibodies for IL-6 or IFNAR1. Scale bar, 50 μm. **c** The density of cells expressing Ctip2, Satb2, Cux1, or Tbr1 per total (DAPI$^+$) cells in 20 μm bins in the CP, which was divided into eight zones (two-sided Steel–Dwass test; $n = 8$ sections from four mice; $^{**}P < 0.01$, $^*P < 0.05$). Data are presented as the mean values ± S.D. The exact $P$ values are summarized in Supplementary Table 1. Source data are provided as a Source Data File.

transcription (STAT) 3 via tyrosine kinase (TYK) 2-, and Janus kinase (JAK) 2-mediated phosphorylation, thereby inducing the transcriptional activation of target genes[38]. Importantly, Satb2 expression has shown to be upregulated cell-extrinsically by ciliary neurotrophic factor and leukemia inhibitory factor, which are members of the gp130 cytokine family, in cervical ganglion neurons undergoing neurotransmitter *trans*-specification[43], suggesting that IL-6, which converges on the gp130 pathway, may enhance the transcription of *Satb2* in CP neurons. In addition, previous studies have revealed that IL-6, via the JAK2/STAT3 signaling pathway, upregulates DNA methyltransferase 1 and downregulates ten–eleven translocation 3, causing the DNA methylation of Neu-roD1 and resulting in the switch from neurogenesis to astrogliogenesis[44,45]. Meanwhile, IFN-I leads to STAT1 and STAT2 activation via TYK2 and JAK1 and induces the expression of IFN-

stimulated genes[39]. IFN-I is also implicated in epigenetic mechanisms that regulate the functions of a wide variety of cell types, including macrophages[46] and pancreatic β cells[47], thereby participating in the initiation of diseases. Such regulation mechanisms mediated by IL-6 and IFN-I might cause the posttranscriptional modification of transcription factors crucial for differentiation of postmigratory neurons.

In summary, microglia, if inadvertently present in the mid-embryonic CP, would disturb the stabilization of the molecular properties of postmigratory neurons via IL-6 and IFN-I they release. Thus, the developing cortex expels microglia from the midembryonic CP to appropriately fine-tune the expression of molecules needed for proper differentiation of postmigratory neurons, thus securing the establishment of functional cortical circuit.

## Methods

**Mice and cell line**. CX3CR1-GFP mice[21] [stock no. 005582] were purchased from Jackson Laboratories (Bar Harbor, ME, USA). $Cxcr4^{-/-}$ mice[26] were introduced from Osaka University. Gadd45g-d4Venus mice[34] [accession no. CDB0490T] were maintained in our laboratory. ICR mice were purchased from Japan SLC (Shizuoka, Japan). To obtain CX3CR1-GFP[+] embryos (heterozygous), male homozygous CX3CR1-GFP mice (8–24 weeks) were mated with female ICR mice (8–24 weeks). The day when the vaginal plug was detected was considered E0. Both male and female embryos (E12–E18) were used for analysis. All mice were maintained under specific pathogen-free conditions, and were housed at 22–24 °C temperature with 40–60% humidity at Nagoya University. A 12 h light/12 h dark cycle was used. The animal experiments were conducted according to the Japanese Act on Welfare and Management of Animals, Guidelines for Proper Conduct of Animal Experiments (published by Science Council of Japan), and Fundamental Guidelines for Proper Conduct of Animal Experiment and Related Activities in Academic Research Institutions (published by Ministry of Education, Culture, Sports, Science and Technology, Japan). All protocols for animal experiments were approved by the Institutional Animal Care and Use Committee of Nagoya University (no. 29006).

Mouse NB2a neuroblastoma cell line was introduced from RIKEN BioResource Research Center [accession no. RCB2639].

**$Cxcr4^{-/-}$ mice genotyping**. PCR-based genotyping was performed using genomic DNA from mouse tails as template. Samples were subjected to PCR amplification with KOD DNA polymerase (Toyobo, Osaka, Japan) for 33 cycles of 10 s at 98 °C, 30 s at 63 °C, and 2 min at 68 °C with preincubation for 2 min at 94 °C. The specific primers used were: 5′-ACG GGG TGA AGT AGG GGA GT-3′ (sense) and 5′-GAG TGT CCA CCC CGC TTT CC-3′ (antisense). PCR products were digested by *XhoI* (Takara, Shiga, Japan) for 2 h at 37 °C, resolved on 1% agarose gels, and visualized by staining with ethidium bromide and UV transillumination.

**Live imaging in cortical slice culture**. To obtain cortical slices covered with intact meninges, whole forebrains isolated from E14 male and female CX3CR1-GFP mice were embedded in 2% agarose gel, and sliced coronally (350 μm) using a vibratome (Microslicer DTK-3000, D.S.K., Kyoto, Japan). The slices were then embedded in type I collagen gel solution (the final concentration: 0.7–0.8 mg/ml; Cellmatrix IA; Nitta Gelatin, Osaka Japan) on a 35 mm glass-bottom (27 mm diameter) dish (IWAKI, Shizuoka, Japan)[48]. When necessary, the meninges were detached from the slices.

Time-lapse imaging data were collected using CV1000 software version 1.06.06 on CellVoyager CV1000 (Yokogawa Electric Corporation). Chambers for on-stage culture were filled with 40% O₂. To characterize their migratory direction, we monitored about 30 cells in each outer or inner region for 8 h and categorized them into three groups according to their migratory behaviors: "basalward," >10 μm toward meninges; "apicalward," >10 μm toward the apical/ventricular surface, and "stationary," ≤10 μm (Figs. 1h, n; 2c, g; and 3i, k; Supplementary Figs. 2d and 8c, g).

**In vivo live imaging using two-photon microscopy**. Pregnant CX3CR1-GFP mice were anesthetized with isoflurane (Cat#099-06571, FUJIFILM Wako Pure Chemical Corp., Osaka, Japan) inhalation. To facilitate the handling of uterine horns, 2 mg per kg ritodrine hydrochloride (Cat#186-02231, FUJIFILM Wako Pure Chemical Corp.), a myometrium relaxant, was intraperitoneally administered. After a midline laparotomy was performed, the distal part of one uterine horn was fully mobilized by cutting the mesometrium and ligating the ovarian vessels (this procedure did not affect uterine circulation or embryos' survival until birth). Then, an embryo (in the uterine tube) was sandwiched by two holding bars equipped in a box, to be subsequently used for further fixing of the embryo with low-melting temperature agarose (5%). The head of the embryo was positioned to make the dorsolateral part of the cerebral hemisphere closest to the objective lens. A coverslip attached to a water container (for immersing the objective lens) was gently pressed to the uterine wall and was tightly held by the subjacent equipment. Throughout the preparative and imaging steps, the mother mouse was warmed using a heater to maintain body temperature. Pallial walls were scanned by an A1RMP two-photon microscopy (Nikon, Tokyo, Japan) using a 20× (N.A. 1.0) water immersion lens (Fig. 1j–o).

**Explant cultures**. The SVZ/VZ and CP explants from E14 and E17 ICR mice, which were manually dissected using microknives, were placed next to an embedded silicone rubber spacer (0.2 mm × 0.2 mm × 3 mm) attached to a culture dish and embedded in collagen gel. After the gel solidified, the rubber spacer was removed to produce a hole, and CX3CR1-GFP[+] microglia harvested from the pallium of E14 CX3CR1-GFP mice were placed in this hole (Fig. 2i–m). These explants were cultured for 3 days. Since the tissue growth rate of the SVZ/VZ explants seemed higher than that of the CP explants (Supplementary Fig. 3b), we performed immunostaining for cleaved caspase-3 (Cl-Casp-3) to detect dying cells to exclude the possibility that the viability of the cells in each explant affects microglial immigration. However, we did not detect a significant difference between the SVZ/VZ and CP explants until 3 days (Supplementary Fig. 3c, d). To test the microglial preferences of explants obtained from E17 cerebral walls, microglia isolated from

E14 brains were cocultured with explants derived from E17 mouse cerebral walls (Supplementary Fig. 3e). Microglia were more reluctant to move toward the tissues, and the density of microglia in the SVZ/VZ explants was comparable with that in the CP explants even 1 day after culturing (Supplementary Fig. 3f, d). Microglia gradually accumulated in the SVZ/VZ explants 3 days after the start of culture, suggesting that microglial attraction toward the SVZ/VZ in this stage was milder compared with that on E14. Moreover, the density of the microglia in the E17 CP was significantly higher than that of microglia in the E14 CP after 1 day and 3 days (Supplementary Fig. 3h), supporting the concept that the CP becomes more accepting of microglial entrance in the later stage.

**Cell sorting**. Freshly isolated pallial walls were treated with trypsin (0.05%, 3 min at 37 °C). Dissociated pallial cells were filtered through a 40-μm strainer (Corning, Corning, NY, USA) to eliminate all remaining cell debris and then resuspended in DMEM containing 5% fetal bovine serum (FBS) (Invitrogen, Waltham, MA, USA), 5% horse serum (HS) (Invitrogen), and penicillin/streptomycin (50 U ml⁻¹, each) (Meiji Seika Pharma Co., Ltd., Tokyo, Japan). CX3CR1-GFP[+] cells or Gadd45g-d4Venus[+] cells were sorted through a 100 μm nozzle using FACSDiva software version 8.0 on FACS SORP Aria II (BD Biosciences, Franklin Lakes, NJ, USA). The drop delay was optimized using BD Biosciences Accudrop™ beads (Cat#345249, BD Biosciences) according to the manufacturer's recommendations. Cerebral wall cells were gated on a forward scatter (FSC)/side scatter (SSC) plot (Fig. 6b; Supplementary Figs. 3a and 14). Debris and dead cells were excluded, and then CX3CR1-GFP[+] or Gadd45g-d4Venus[+] cells were further gated for sorting. For RNA-Seq and FACS analyses, for which we needed to prepare many microglia, CD11b[+] cells were collected using a magnetic bead separation (MACS) system (Cat#130–093–634, Miltenyi Biotec, Bergisch Gladbach, Germany) from ICR mouse cerebral wall cells.

**Administration of AMD3100 into the mouse ventricle**. After pregnant ICR mice were anesthetized by intraperitoneal injection of pentobarbital sodium, somnopentyl (Kyoritsu Seiyaku, Tokyo, Japan), AMD3100 (Cat#A5602-5MG, Sigma-Aldrich, St. Louis, MO, USA) dissolved in saline was injected into the lateral ventricle of embryos at E14 (22.5 μg). After 24 h, the brains of embryos (E15) were perfused with 4% paraformaldehyde (PFA), and were subjected to immunohistochemistry.

**Microglial attraction by CXCL12-soaked beads**. Brains of E15 CX3CR1 mice or $Cxcr4^{-/-}$ mice were embedded in 2% agarose gel, and then sliced coronally (400 μm thick). Slices were placed on Millicell membrane inserts (0.4 μm) (Merck Millipore, Darmstadt, Germany) plated on 35 mm plastic dishes (Corning) to expose the upper surface of the slice to a humidified atmosphere containing 40% O₂ and 5% CO₂. Resin beads (Cat#143–2446, AG 1–X8 resin, Bio-Rad, Hercules, CA, USA) soaked in 1 μg ml⁻¹ CXCL12 (Cat#460–SD–010, R&D Systems, Minneapolis, MN, USA) were deposited on the basal surface of the brain slices. The number of CXCL12 beads used per sliced hemisphere (one side) was ~40–60. After culture for 24 h at 37 °C, the slices were fixed in 4% PFA for immunohistochemistry (Fig. 4a–g; Supplementary Fig. 9). Normally, the ratio of microglia to nonmicroglial cells (other cerebral wall cells) is about 0.5–1:100 in E14 cortex, but our live observation revealed that microglia can extensively survey the cerebral wall and interact with many surrounding cells[12]. Considering this surveillance range of microglia, our microglial delivery into the CP should have provided a sufficient density of microglia to verify their effect on the CP.

To test whether the attraction for microglia to the CP was not due to abnormal cell death induced by beads placed on slices (because microglia are known to react to tissue damage), we performed Cl-Casp-3 immunostaining to detect dying cells. There were few Cl-Casp-3[+] cells in each condition after 24 h of culture, showing that CXCL12 beads specifically recruited microglia into the CP and that the cultured slices were healthy (Supplementary Fig. 11). In addition, we did not find microglial accumulation in the CP in $Cxcr4^{-/-}$ slices on which CXCL12 beads were placed. In this case, the density of cells positive for each neuronal layer marker was comparable overall (Supplementary Fig. 9).

**Microglial transplantation**. CX3CR1-GFP[+] microglia were harvested from E15 CX3CR1-GFP mice using a cell sorter (FACS SORP Aria II) and then suspended in saline at a density of $1.0 \times 10^4$ cells μl⁻¹. The cells were transplanted into the CP by injection through a glass capillary into isolated E15 ICR brain from the backside of the forebrain, just beneath the meninges; the capillary was pulled backward while the cells were ejected. For the control, only saline was injected into the brains. After transplantation, brains were embedded in 2% agarose gel and then sliced coronally (400 μm thick). The slices were cultured on Millicell membrane inserts (0.4 μm) (Merck Millipore) placed on a 35 mm culture dish for 24 h at 37 °C, and then fixed and subjected to immunohistochemistry (Supplementary Fig. 12).

**In vivo CXCL12 overexpression in postmigratory neurons**. To overexpress CXCL12 in neurons in the CP to timely recruit microglia to the CP on E15–E16, we used a tamoxifen-inducible form of Cre recombinase (Cre-ER) under the control of a neuron-specific Tα1-promoter to initiate the expression of EF-LPL-Lyn-TdTomato (used as an indicator) and/or EF-LPL-CXCL12. These vectors were

introduced by IUE on E12 and were followed by maternal tamoxifen administration on E14. As we know that microglia respond to plasmid DNA injected into the ventricle via microglia-expressing TLR9 recognition and accumulate near the apical surface[49], we coadministered a TLR9 antagonist, 0.5 µg of oligonucleotide (ODN) 2088 (Cat#tlrl-2088, InvivoGen), with plasmid DNA to prevent unnecessary microglial activation (Figs. 4h–l and 10). To exclude the possibility that CXCL12 can directly affect the expression of neuronal marker genes independent of microglia, we evaluated mice intraventricularly administered clodronate (Sigma-Aldrich), which selectively depletes microglia, at the time of tamoxifen induction (on E14). For neutralization of IL-6 and IFN-I in vivo, CX3CR1-GFP mice were intraventricularly administered InVivoMab anti-mouse IFNAR1 (Cat#BE0241, BioXCell, West Lebanon, NH, USA) and/or InVivoMab anti-mouse IL-6 (Cat#BE0046, BioXCell) (20 µg) on E14 and then analyzed on E16 (Fig. 10).

**Immunofluorescence**. Brains were fixed in 4% PFA, immersed in 20% sucrose, and then frozen sectioned (16 µm). Cultured cells were fixed in 4% PFA for 10 min at room temperature, washed in PBS, and then immunostained. Sections or cells were treated with the following primary antibodies: mouse anti-βIII-tubulin mAb (1:1000, Cat#MMS-435P, Covance, Princeton, NJ, USA); rabbit anti-βIII-tubulin pAb (1:1000, Cat#802001, BioLegend, San Diego, CA, USA); rabbit anti-Cl-Casp-3 pAb (1:500, Cat#9661s, Cell Signaling Technology Inc., Beverly, MA, USA); goat anti-CD206 pAb (1:200, Cat#AF2535, R&D systems); rat anti-Ctip2 mAb (1:1000, Cat#ab18465, Abcam, Cambridge, UK); rabbit anti-Cux1 pAb (1:200, Cat#sc-13024, Santa Cruz Biotechnology, Inc., Santa Cruz, CA, USA); chicken anti-GFP pAb (1:1000, Cat#GFP-1020, Aves Labs, Tigard, OR, USA); rat anti-GFP mAb (1:500, Cat#GF090R, Nacalai Tesque, Kyoto, Japan); rabbit anti-Iba1 pAb (1:2000, Cat#019–19741, FUJIFILM Wako Pure Chemical Corp.); mouse anti-MAP2 mAb (1:5000, Cat#M1406, Sigma-Aldrich); rabbit anti-P2RY12 pAb (1:500, Cat#AS–55043A, AnaSpec, San Jose, CA, USA); rabbit anti-RFP pAb (1:1000, Cat#PM005, MBL);mouse anti-Satb2 mAb (1:400, Cat#ab51502, Abcam), rabbit anti-Tbr1 pAb (1:500, Cat#ab31940, Abcam), and rabbit anti-Tbr2 mAb (1:500, Cat#ab183991, Abcam). After washes, sections were treated with secondary antibodies conjugated to Alexa Fluor 488, Alexa Fluor 546, or Alexa Fluor 647 (1:500, Cat#A10040, A11029, A11030, A11039, A11055, A11056, A11081, A21202, A21206, A21208, A21245, A21247, A21447, Invitrogen), and then were stained with DAPI (Cat#D9542, Sigma-Aldrich). After staining, the sections were mounted with mounting solution. When necessary, antigens retrieval was performed by heating samples at 70 °C in HistoVT One (Cat#06380–05, Nacalai tesque) for 20 min. Image data were collected using FV10–ASW software version 4.1 on Fluoview FV1000 (Olympus) and NIS-Elements software AR Analysis version 5.01.00 on TiEA1R (Nikon) and A1Rsi (Nikon).

**Coculture of microglia with cortical neurons**. Cerebral walls were dissected from E14 Gadd45g-d4Venus mice and dissociated into single cells. Gadd45g-d4Venus+ cells were isolated by FACS, and seeded at a density of $3.3 \times 10^5$ cells per cm$^2$ on glass coverslips (5 mm diameter, Matsunami Glass, Osaka, Japan) coated with polyethylenimine (Sigma-Aldrich). Each coverslip was placed in a compartment formed by a silicone rubber ring (8 mm inner diameter, 1.5 mm thick) on a 35 mm plastic dish. After cells adhere to the cover glass, culture media were added to 100 µl to fill the compartment. Use of a silicone rubber-based chamber was necessary to prevent the culture medium from drying up. Culture was performed in DMEM supplemented with 5% FBS, 5% HS, 50 U ml$^{-1}$ penicillin/streptomycin, 10 ng ml$^{-1}$ EGF (PeproTech, Inc., Rocky Hill, NJ, USA), and 10 ng ml$^{-1}$ bFGF (Invitrogen) in a 5% CO$_2$/37 °C incubator.

After Gadd45g-d4Venus+ cells on coverslips were cultured for 2 days to allow them to differentiate into CP-like (d4Venus−) neurons, separately sorted CX3CR1-GFP+ cells were added to the neuronal cultures (at a density of $8.3 \times 10^4$ cells per cm$^2$ microglia with a 1:0.25 neuron:microglia ratio per coverslip). After 24 h, immunocytochemistry was performed (Fig. 5a–f). In vivo, microglia were almost absent from the CP until E17, with first microglial reentrance into the deep part of the CP (future layers 6 and 5) occurring on E17–E18[14]; this left the more superficial CP microglia-free until postnatal day 2 (P2)–P3. Therefore, our protocol of coculturing 2 days in vitro (d.i.v.) neurons harvested from E14 Gadd45g-d4Venus mice with microglia (for an additional 24 h until 3 d.i.v.) allowed immature CP-like in vitro neurons, corresponding to neurons in the in vivo CP on E16–17, to artificially encounter excessive microglia. To neutralize IL-6 and IFN-I released from microglia, microglia were pretreated with anti-mouse IL-6-IgG (Cat#mabg-mil6-3, InvivoGen, San Diego, CA, USA) or anti-mouse IFNAR1 (Cat#BE0241, BioXCell) for 30 min, and then culture media containing both microglia and antibodies were added to the 2 d.i.v. CP-like neuron cultures at a concentration of 5 µg ml$^{-1}$ (Fig. 9c, d).

**RNA-Seq analysis**. After being cocultured with CD11b+ microglia for 24 h, in vitro-prepared CP neurons were purified through CD11b depletion using the MACS system. Total RNA from these neurons was extracted using the RNeasy Micro Kit (Cat#74004, Qiagen, Hilden, Germany). Libraries were prepared using the TruSeq Stranded mRNA LT Sample Prep Kit (Illumina, San Diego, CA, USA) and then were further sequenced on the Illumina NovaSeq6000 platform using a 100 bp paired-end strategy at Macrogen (Seoul, South Korea). Read qualities were

assessed by the FASTQC tool on Galaxy. Reads were then mapped to the mouse genome assembly (mm10) using TopHat version 2.1.1 using the corresponding sample's mean inner distance between mate pairs. mRNA read counts were quantified during transcript assembly with Cufflinks version 2.2.1.2. For individual gene plots, we calculated fragments per kilobase of exon per million mapped reads (FPKM).

To exclude the possibility that the samples were contaminated with microglia or perivascular macrophages, we investigated the expression levels of several markers specific for microglia and macrophages (Fig. 7a). For the heat map, the count data were transformed using the DESeq2 algorithm (Fig. 7b; Supplementary Fig. 16). For DEG analysis, we assessed only genes that were reliably detected in both groups (11,603 genes). Based on the statistical threshold (log$_2$ FC > 1.4, FDR < 0.1), we identified 143 downregulated genes and 246 upregulated genes in the neurons$^{MG}$ group (Fig. 8a–c). Seeking an unbiased approach to pathway analysis, we used the GSEA tool developed by Broad Institute (http://software.broadinstitute.org/gsea/index.jsp), which identifies groups of coordinately regulated genes present in gene sets annotated in the Molecular Signatures Database (MSigDB) (Fig. 8d, e). Only genes with an FPKM of >1 were used for GSEA in all replicates of any one condition. The FDR for GSEA is the estimated probability that a gene set with a given normalized enrichment score represents a false-positive finding, and an FDR < 0.25 is considered statistically significant for GSEA. The RNA-Seq dataset is available at Supplementary Data 1. The raw data have been deposited in the DNA Data Bank of Japan (DDBJ) under the DRX accession number: DRX199371–DRX199376.

**Intracellular staining for FACS analyses**. After being cocultured with CD11b+ microglia for 24 h, in vitro-prepared CP neurons were isolated through CD11b depletion with MACS beads. Intracellular staining was performed using the Transcription Factor Buffer Set (Cat#562574, BD Biosciences) according to the manufacturer's recommendations. To block nonspecific antibody binding, cells were treated with an anti-mouse CD16/CD32 antibody (1:400, Cat#553141, BD Biosciences) for 20 min at 4 °C before staining. The cells were treated for 60 min at 4 °C with the following primary antibodies: goat anti-Brn2 pAb (1:100, Cat#sc-6029, Santa Cruz Biotechnology); rat anti-Ctip2 mAb (1:1000, Cat#ab18465, Abcam); mouse anti-Cux1 mAb (1 µg ml$^{-1}$, Cat#ab54583, Abcam); rabbit anti-RORβ pAb (1 µg ml$^{-1}$, Cat#LS-A2374, LifeSpan BioSciences, Seattle, WA, USA); mouse anti-Satb2 mAb (1:400, Cat#ab51502, Abcam); rabbit anti-Tbr1 pAb (1:500, Cat#ab31940, Abcam), mouse IgG1 isotype control (1:400, Cat#M075–3, MBL), rat IgG2a isotype control (1:400, Cat#553988, BD Biosciences); goat IgG control (1 µg ml$^{-1}$, Cat#15256–10MG, Sigma-Aldrich), and rabbit IgG control (1 µg ml$^{-1}$, Cat#02–6102, Invitrogen). After washing, the cells were treated with Donkey anti-Mouse IgG secondary antibody conjugated with Alexa Fluor 488 (1:500, Cat#A21202, Invitrogen), Donkey anti-goat IgG secondary antibody conjugated with Alexa Fluor 647 (1:500, Cat#21447, Invitrogen), and Brilliant Violet 421 Donkey anti-rabbit IgG antibody (1:400, Cat#406410, BioLegend) for 30 min at 4 °C. Data collection was performed using FACSDiva software version 8.0 on FACS Canto II (BD Biosciences) and the data were analyzed using FlowJo software version 7.6. About 5000 cells (Fig. 6b–d) or 3000 cells (Fig. 9d, e; Supplementary Figs. 17 and 18), which were gated on the FSC/SSC plot for debris exclusion, were analyzed for each sample.

**RT-PCR**. The first-strand cDNA was synthesized from ~0.1 µg of total RNA using SuperScript III (Invitrogen). To amplify specific transcripts, samples were subjected to PCR for 32 cycles of 10 s at 98 °C, 30 s at 60 °C, and 1 min at 68 °C with preincubation for 2 min at 94 °C using KOD Fx Neo DNA polymerase (Toyobo). The primers used were 5′-GGA ACC GAT CAG TGT GAG TAT-3′ (sense) and 5′-ACA GGC TAT CGG GGT AAA GG-3′ (antisense) for mouse Cxcr4 and 5′-GTT GTC TCC TGC GAC TTC A-3′ (sense) and 5′-GGT GGT CCA GGG TTT CTT A-3′ (antisense) for mouse glyceraldehyde-3-phosphate dehydrogenase (Gapdh). PCR products were resolved on 1% agarose gels and visualized by staining with ethidium bromide and UV transillumination.

**In situ hybridization**. Sense and antisense RNA probes for CXCL12 were prepared from E14 mouse cortex cDNA using a digoxigenin (DIG) RNA labeling kit (Cat#11175025910, Roche, Basel, Switzerland). Brains were fixed overnight in 4% PFA, and 16-µm-thick cryosections were hybridized with a volume of 200 µl of hybridization buffer (5x SSPE, 0.1% SDS, 640 µg ml$^{-1}$ yeast t-RNA, and 50% deionized formamide) containing DIG-labeled RNA probes (500 ng ml$^{-1}$)[50]. RNA probes in hybridization buffer were pretreated at 85 °C for 5 min and then placed on the cryosections. After overnight hybridization at 65 °C, the sections were rinsed and stained with nitro blue tetrazolium chloride (Cat#N5514-10TAB, Sigma-Aldrich)/5-bromo-4-chloro-3 indolyl phosphate (Cat#B0274-10TAB, Sigma-Aldrich) solution in a dark room. The following day, the sections were washed and then mounted with mounting solution for observation under a microscope. Sequences of primers used to synthesize probes for Cxcl12 are shown as follows: 5′-ACA GCA CTG ACT GGG GTC AT-3′ (sense) and 5′-ACT AAT ACG ACT CAC TAT AGG GAC TGT GGC TTC ATG GCA AGAT-3′ (antisense) for naive Cxcl12 expression (Fig. 3a; Supplementary Fig. 4a) and 5′-GCA ATT GTT GTT GTT AAC TTG-3′ (sense) and 5′-ACT

AAT ACG ACT CAC TAT AGG GGG AAT GGA CAG CAG GGG GCT-3′ (antisense) for *Cxcl12* overexpression (Fig. 4i).

**ELISA**. The production of mouse IL-6, IFN-α, and IFN-β was measured in the supernatants of cocultured neurons and/or microglia using a Mouse IL-6 Quantikine ELISA Kit (Cat#M6000B, R&D Systems), IFN-α Mouse ELISA Kit (Cat#BMS6027, Thermo Fisher Scientific), and Mouse IFN-β Quantikine ELISA Kit (Cat#MIFNB0, R&D Systems), respectively. The optical density (OD) of each well was determined at 450 and 570 nm using a PowerScan4 plate reader (BioTek, Winooski, VT, USA). The background-adjusted OD for each sample was calculated by subtracting the OD (570 nm) reading from the OD (450 nm) reading.

**Statistics and reproducibility**. Quantitative data are presented as the mean values ± S.D. of representative experiments. Statistical differences between groups were analyzed using the Mann–Whitney $U$ test for two-group comparisons, the Steel–Dwass test for multiple comparisons and Pearson's chi-squared test for contingency tables evaluating microglial migration patterns on R software version 3.6.0. All statistical tests were two-tailed and $P < 0.05$ was considered significant. For RNA-Seq, FDR value was calculated from the two-tailed $P$ values using the R software package qvalue, and the gene with FDR < 0.1 was considered to be significantly different between groups (Fig. 7b; Supplementary Fig. 16). Individual values were plotted as circles in bar graphs. The number of samples examined in each analysis is shown in the corresponding figure legend. No randomization was used. Mice analyzed were litter mates whenever possible. No samples were basically excluded from the analysis. We only excluded the data obtained from failed experiments by some reasons, e.g., failure in sample preparations. No statistical methods were used to predetermine the sample size owing to experimental limitations. Sample size was determined to be adequate based on the magnitude and consistency of measurable differences between groups. Basically, investigators were blinded during experiments. We confirmed that replicate experiments were successful by repeating at least three times for all experiments. The micrographs in the figures are shown as the representative.

**Reporting summary**. Further information on research design is available in the Nature Research Reporting Summary linked to this article.

## Data availability
The source data underlying Figs. 1b, d, g–i, n, o, 2b–d, f–h, m, 3c–e, g–l, 4c, d, g, l, 5f, i, 6d, 7, 9a, e, and 10c, and Supplementary Figs. 2c, d, 3b, d, g, h, 4b, c, 5b–d, f–h, 6b, c, 7a–d, 8a–h, 9b, 10a, b, 11b, 12c, d, f, 13, 15, 16, 17b, and 18b are provided as a Source Data File. The RNA-Seq dataset is available at Supplementary Data 1. The raw data have been deposited in the DNA DDBJ under the DRX accession number: DRX199371–DRX199376. These sequence data are also available at NCBI Sequence Read Archive under the ID code: DRP005827. For all other inquiries, please contact the corresponding authors.

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

## Acknowledgements

We thank Makoto Masaoka and Namiko Noguchi (Department of Anatomy and Cell Biology, Graduate School of Medicine, Nagoya University) for technical assistance. We are grateful to Carina Hanashima (Department of Biology, Faculty of Education and Integrated Arts and Sciences, Waseda University) and Koji Oishi (Department of Anatomy, Keio University School of Medicine) for helpful advice and suggestion. This work was supported by JSPS KAKENHI Grant numbers JP16H02457 [T.M.], JP16K15169 [T.M.], JP16J06207 (Grant-in-Aid for JSPS Fellows) [Y.H.] and JP18K15003 (Grant-in-Aid for Young Scientists) [Y.H.], and by a grant from the Uehara Memorial Foundation (Grant number 201910147) to Y.H. The two-photon imaging in this study was supported by NIBB Collaborative Research Program for Integrative Imaging (16-504) to T.M.

## Author contributions

Y.H. and T.M. designed the study and wrote the paper. Y.H. performed most of the experiments and data analysis. Y.N. assisted with data collection for live imaging and immunohistochemical analyses. Y.T. analyzed the RNA-Seq data. S.N. and H.W. supported in utero observation using two-photon microscopy. T.N. provided *Cxcr4*$^{-/-}$ mice. S.N., H.W., T.N., and A.K. contributed to interpretation and critically reviewed the manuscript. Y.H. created Figs. 1j, k, 2j, 3g, i, k, 4a, h, 5b, g, 6a, 9c, and 10a, and Supplementary Figs. 2a, b, 12a, and 19. All authors approved the final version of the manuscript, and agree to be accountable for all aspects of the work.

## Competing interests

The authors declare no competing interests.
