## [Peer Review File · Nature Communications]

REVIEWERS' COMMENTS:

Reviewer #1 (Remarks to the Author):

The authors completed much additional work for this revision and, most importantly, softened the claims about a change in neuronal identity. This is important because the manuscript does not prove that the subtype identity of the neurons are changed due to prolonged microglia in the cortical plate. Using bulk sequencing of in-vitro neurons, the authors claim that “abnormal exposure to microglia widely disturbed the expression of subtype-associated transcription factors and reduced the expression of molecules involved in the functional maturation of cortical neurons”. Largely, their bulk sequencing results demonstrate this; however, there are glaring inconsistencies (i.e. Bcl11b), as the authors note, from their immunocytochemistry analyses from Figure 5. It is likely that the inconsistencies may come from the fact that bulk seq included a large variety of cell types. While the authors may not have access to advanced sequencing methods (e.g. single cell sequencing), they should have sorted subtypes of neurons (already done in Figure 6), for “subtype-specific” bulk-sequencing. This would have provided a cleaner, more reliable data set on the alteration to “neuronal and functional differentiation” that is claimed. My suggestion for solving this and allow timely publication would be that of removing any claim on subtype identity as the data have inconsistencies in that regard.

Reviewer #2 (Remarks to the Author):

The authors have addressed all my concerns regarding the figures and experimental details.

Thank you so much for reviewing our manuscript entitled "**Transient microglial absence assists postmigratory cortical neurons in proper differentiation**" at *Nature Communications*. We would like to describe our response to the given comments in point-by-point manner as below. Changes to the manuscript in accordance with Reviewer 1's comment are **highlighted in yellow**.

Reviewer #1 (Remarks to the Author):

The authors completed much additional work for this revision and, most importantly, softened the claims about a change in neuronal identity. This is important because the manuscript does not prove that the subtype identity of the neurons are changed due to prolonged microglia in the cortical plate. Using bulk sequencing of in-vitro neurons, the authors claim that "abnormal exposure to microglia widely disturbed the expression of subtype-associated transcription factors and reduced the expression of molecules involved in the functional maturation of cortical neurons". Largely, their bulk sequencing results demonstrate this; however, there are glaring inconsistencies (i.e. Bcl11b), as the authors note, from their immunocytochemistry analyses from Figure 5. It is likely that the inconsistencies may come from the fact that bulk seq included a large variety of cell types. While the authors may not have access to advanced sequencing methods (e.g. single cell sequencing), they should have sorted subtypes of neurons (already done in Figure 6), for "subtype-specific" bulk-sequencing. This would have provided a cleaner, more reliable data set on the alteration to "neuronal and functional differentiation" that is claimed. My suggestion for solving this and allow timely publication would be that of removing any claim on subtype identity as the data have inconsistencies in that regard.

We thank the reviewer for positively evaluating our revised manuscript and her/his understanding. We softened the claim throughout the manuscript in the previous version, but in this final version we further carefully reviewed our descriptions/statements throughout the manuscript and removed any claim on "subtype identity" or the related expression which might mislead the readers.

First, in Summary section, we paid more attention and modified some phrases to make it clear that microglia affect neuronal proper "differentiation". In the previous version, we used the sentence which might be suggestive of "subtype identity", but we revised the text as follows.

Line 43–44 of page 2 (Summary)

Upon nonphysiological excessive exposure to microglia *in vivo* or *in vitro*, young postmigratory and *in vitro*-grown CP neurons **showed abnormal differentiation with disturbed expression of the subtype-associated transcription factors and genes implicated in functional neuronal maturation.**

Second, we revised Discussion section. Originally, we described in detail about the possible mechanism underlying the expression pattern of neuronal subtype-associated genes (*Satb2*, *Tbr1*, *Cux1*, *Fezf2* and *Sox5*), but we removed this part with some concerns that these descriptions might help the readers remind of “subtype identity” and as a result would confuse them. This removal was also done to reduce the word number within 5,000 words.

Third, in Result section, we tried to exactly explain the results and the inconsistency between our immunostaining and RNA-Seq analyses.

We minorly revised the order of descriptions on results and added some text regarding *Tbr1* expression as follows.

Line 266–274 of page 13 (Results, section 4)

Heat map data revealed that most of UL marker genes (*Satb2*, *Mdga1*, *Frmd4b*, *Cux2*, *Btg1* and *Inhba*) were more strongly expressed in neurons^{MG} than in neurons^{Cont} (Fig. 7c; Supplementary Fig. 16; Supplementary Table 1), although there were some exceptions; the expression of *Plxnd1* and *Lpl*, which are known as L2/3 and L5 marker, was decreased. Meanwhile, the expression of many of DL marker genes (*Tle4*, *Lmo3*, *Foxo1*, *Nr4a3*, *Grb14*, *Sox5*, *Sla*, *Lxn*, *Cdh13* and *Pcp4*) was decreased. These data showed a tendency to reduce the expression of DL marker genes and increase the expression of typical UL marker genes. In addition, the expression of *Tbr1* was increased in neurons^{MG} compared to that in neurons^{Cont}, which is consistent with the results obtained from the immunohistochemical analyses.

We tried to explain more deliberately the reason for the difference between our immunostaining and RNA-Seq analysis.

Line 275–282 of page 13 (Results, section 4)

The aforementioned immunocytochemical data showed that the proportion of *Ctip2*⁺ in neurons^{MG} was decreased, but our RNA-Seq analysis revealed that the expression of *Bcl11b* (*Ctip2*) was conversely increased. In addition, the expression of *Cux1*, whose expression was increased in immunocytochemical analysis, in neurons^{MG} was comparable to neurons^{Cont} in RNA-Seq analysis. These inconsistencies would reflect that bulk RNA-Seq, even performed for neurons enriched based on culture from the same SVZ region, may have included a large variety of cell types, whereas immunocytochemistry had a single-cell resolution. Or, they might be explained by the possible effects of microglia on the posttranscriptional regulation of neurons.

In addition, we corrected some misleading words which are suggestive of “subtype identity” as follows.

Line 211–212 of page 10 (Results, section 3)

These results obtained by slice culture-based and in vivo experiments suggest that microglia may influence

postmigratory cortical neurons in the expression of neuronal subtype-associated transcription factors, ...

Line 396–400 of page 18–19 (Discussion)

Our molecular characterization coupled with the artificial/experimental exposure of CP neurons to microglia revealed that disturbances in the subtype-associated gene expression were primarily mediated by IL-6 and IFN-I. However, the detailed mechanism by which these cytokines modulate the expression of transcription factors fundamental for neuronal proper differentiation remains unresolved.

We hope this version is now suitable for publication.

Reviewer #2 (Remarks to the Author):

The authors have addressed all my concerns regarding the figures and experimental details.

We thank the reviewer for the positive comment.

We would like to thank again the editor and all reviewers for providing constructive reviews.